# Reward Engineering for Spatial Epidemic Simulations: A Reinforcement Learning Platform for Individual Behavioral Learning

**Radman Rakhshandehroo**                                    *rdmnr@student.ubc.ca*
*Department of Computer Science*
*University of British Columbia*

**Daniel Coombs**                                            *coombs@math.ubc.ca*
*Department of Mathematics and Institute of Applied Mathematics*
*University of British Columbia*

**Reviewed on OpenReview:** *https: // openreview. net/ forum? id= yPEASsx3hk*

## Abstract

We present ContagionRL, a Gymnasium-compatible reinforcement learning platform specifically designed for systematic reward engineering in spatial epidemic simulations. Unlike traditional agent-based models that rely on fixed behavioral rules, our platform enables rigorous evaluation of how reward function design affects learned survival strategies across diverse epidemic scenarios. ContagionRL integrates a spatial SIRS+D epidemiological model with configurable environmental parameters, allowing researchers to stress-test reward functions under varying conditions including limited observability, different movement patterns, and heterogeneous population dynamics. We evaluate five distinct reward designs, ranging from sparse survival bonuses to a novel potential field approach, across multiple RL algorithms (PPO, SAC, A2C). Through systematic ablation studies, we identify that directional guidance and explicit adherence incentives are critical components for robust policy learning. Our comprehensive evaluation across varying infection rates, grid sizes, visibility constraints, and movement patterns reveals that reward function choice dramatically impacts agent behavior and survival outcomes. Agents trained with our potential field reward consistently achieve superior performance, learning maximal adherence to non-pharmaceutical interventions while developing sophisticated spatial avoidance strategies. The platform's modular design enables systematic exploration of reward-behavior relationships, addressing a knowledge gap in models of this type where reward engineering has received limited attention. ContagionRL is an effective platform for studying adaptive behavioral responses in epidemic contexts and highlight the importance of reward design, information structure, and environmental predictability in learning. Our code is publicly available at `https://github.com/redradman/ContagionRL`.

## 1 Introduction

The dynamics of infectious outbreaks, epidemics or pandemics, are intricately linked to individual behavior (Wang et al., 2021; Funk et al., 2010). Behavioral decisions such as movements, risk perceptions and adherence to non-pharmaceutical interventions (such as mask-wearing and social distancing) could heavily affect how an epidemic unfolds (Wang et al., 2021; Funk et al., 2010; Manrubia & Zanette, 2022; Ye et al., 2020; Saad-Roy & Traulsen, 2023). Traditional epidemic models primarily use ordinary differential equations (ODEs) to keep track of disease population-wide with models such as SIR (Kermack & McKendrick, 1927) or SEIR (Kermack & McKendrick, 1927; Wilson & Worcester, 1945). While these models are effective for a high-level overview of the disease outbreak, due to their assumption of homogeneity, they neglect nuanced effects of individual behavior and spatial interactions (Bansal et al., 2007; Zachreson et al., 2022). Furthermore, they

are incapable of capturing dynamic behavioral changes or adaptive interventions (Funk et al., 2010; Manrubia & Zanette, 2022; Bansal et al., 2007).

The SARS-CoV-2 pandemic highlighted how individual-level decisions and behaviors, such as adherence to non-pharmaceutical interventions (NPIs) like mask wearing, could profoundly change the trajectory of a disease outbreak (Saad-Roy & Traulsen, 2023). Agent-Based Models (ABMs) (Grimm & Railsback, 2013), also known as Individual-Based Models (IBMs), offer a compelling alternative by simulating systems from the bottom up, representing individual agents with specific attributes, spatial contexts, and behavioral rules. These models accommodate heterogeneity and complex interactions, thereby making ABMs particularly suitable for studying epidemics where individual actions drive epidemic-scale outcomes, while leaving them reliant on prescribed or rule-based behaviors rather than learned policies that exhibit more complexity and are more dynamic (Grimm & Railsback, 2013).

The lack of the learned policies could be addressed with Reinforcement Learning (RL) (Sutton & Barto, 2018) and Deep Reinforcement Learning (DRL) (Li, 2017; Mousavi et al., 2016; François-Lavet et al., 2018) that provide powerful frameworks to optimize sequential decision making in dynamic systems (Mnih et al., 2015; Silver et al., 2016; Arulkumaran et al., 2017) by modeling them as Markovian decision processes (MDP). Such algorithms have been applied in epidemiological contexts to learn the optimal policies by focusing on centralized control of macroscopic interventions (Libin et al., 2021; Kwak et al., 2021; Du et al., 2023; Wan et al., 2021). Combining of RL or DRL algorithms with ABMs allows simulation of both adaptive and heterogeneous behaviors.

However, the main challenge in creating and developing informative ABMs lies in defining realistic and computationally tractable agent behaviors. Furthermore, RL-based approaches are highly sensitive to the reward function and the reward design can have considerable impact on the learned policy and the downstream interpretations of the model upon behavioral convergence (Ng et al., 1999; Ibrahim et al., 2024; Nguyen et al., 2025; Shihab et al., 2025).

Currently, there are limited simulation platforms capable of supporting the integration of RL that permit the agents to learn complex strategies for navigation and adherence strategies by direct interaction within their environment. This creates an important gap in the tools for investigating impact of reward design on learned dynamic behaviors at the individual level within biologically sound epidemic simulations.

To address this, we introduce CONTAGIONRL, a specialized reinforcement learning platform designed to study how different reward structures influence agent behavior during epidemic simulations. This is a novel and parameterizable, Gymnasium compatible environment that unifies spatial compartmental epidemic modeling and reinforcement learning. We integrate an extension of a Susceptible-Infected-Recovered-Susceptible (SIRS) framework called SIRS+D, which has an additional component for death, to track population-wide epidemiological dynamics, while also enabling the use of RL to model individual-level decision-making. This environment simulates a single learning agent among a population of non-learning humans, whose movements affect the disease outbreak. In this setting, we frame the agent's decision making as a sequential control problem in which the agent learns behavioral policies through direct interaction with the environment. These policies include both spatial navigation and adherence to non-pharmaceutical interventions (NPIs), developed in response to localized epidemic conditions. Table 4 compares our models to other approaches.

To demonstrate the effectiveness of this novel environment, we conduct a variety of experiments. We investigate the nature of the policies learned by this RL agent under varying epidemiological and behavioral conditions. In addition, we conduct ablation studies to understand the critical components of the reward function that is the most impactful in the learning process. Benchmarking shows that these learned policies significantly outperform random and stationary baselines in terms of survival time. An ablation study shows the importance of both directional cues and adherence incentives within the reward function to learn a robust, risk-averse behavior. Across all effective reward structures, the agent consistently learned maximal adherence to NPIs. This framework provides a modular and highly configurable environment for studying emergent behaviors at the individual level within spatial epidemic models.

## 2 Related Works

**Compartmental Models in Epidemiology**. Classical epidemiological models, such as SIR (Susceptible-Infected-Recovered), divide a population into discrete groups (compartments) based on disease state (Kermack & McKendrick, 1927; Wilson & Worcester, 1945). The flow of individuals between compartments is described by ordinary differential equations (ODEs). Extensions like SEIR (adding an exposed group to act as a disease incubation period) or SVE(R)IRS (adding Vaccinated, accounting for waning immunity) incorporate more states to increase realism (Brauer et al., 2019; Friedman & Kao, 2014; Hethcote, 2000; Chyba et al., 2024; Balisacan et al., 2021). While traditional compartmental models provide valuable population-level insights, assumptions of homogeneity limit their ability to represent individual decision-making (Cangiotti et al., 2024; Siegenfeld et al., 2022; Dimarco et al., 2021; Jiang et al., 2024; Bostanci & Conrad, 2025; Rose et al., 2021; Kong et al., 2016). These models also struggle to capture dynamic behavioral changes or adaptive interventions (Brauer et al., 2019; Siegenfeld et al., 2022). Ignoring such factors can limit accuracy and lead to misleading conclusions (Sudhakar et al., 2024; Siegenfeld et al., 2022). Stochastic formulations are particularly relevant for small populations or during the early stages of an outbreak (Brauer et al., 2019).

**Agent-Based Models (ABMs)**. ABMs address the limitations of compartmental models like SIR by modeling heterogeneous individuals using specific attributes and interaction rules within an environment or network (Tracy et al., 2018; Hunter et al., 2018; 2017). Furthermore, ABMs excel at capturing individual variability, stochasticity, local interactions, and complex contact networks, which are difficult for compartmental models (Siegenfeld et al., 2022; Hunter et al., 2017). Recent advances demonstrate substantial sophistication such as large-scale ABMs that can simulate millions of agents with GPU acceleration (Chopra et al., 2021) or employ LLM-based adaptive agents (Park et al., 2023).

**Challenge of Reward Engineering**. Sophisticated ABMs show potential for learning-enabled epidemic modeling. However, systematic evaluation of reward function design remains an important methodological gap. In reinforcement learning, reward design shapes the fundamentals of the learned behavior with different reward structures leading to considerably different learned policies at convergence (Ng et al., 1999; Amodei et al., 2016; Nguyen et al., 2025; Shihab et al., 2025). Despite the importance of this problem in RL, epidemic modeling applications have not systematically investigated how reward structure affects learning under varying conditions.

**Reinforcement Learning (RL) for Epidemics**. RL has been increasingly leveraged to design policies for epidemic intervention at population and individual level. Individual-based Reinforcement Learning Epidemic Control Agent (IDRLECA) proposed an RL agent at an individual level using GNNs to estimate infection probabilities and balance epidemic suppression vs cost of mobility in its reward function (Feng et al., 2023). VEHICLE (Feng et al., 2022) handles challenges of unobservable asymptomatic cases and delayed effects by using a combination of hierarchical RL with GNNs. Recent work has also focused on using actor critic methods to minimize a cost that considers both the epidemiological, economic and social cost for multi-intervention planning (Mai et al., 2023) or using RL for scheduling lockdown periods (Arango & Pelov, 2020) that incorporates the ICU capacity and minimizing time spent in lockdowns. HRL4EC (Du et al., 2023) also leverages hierarchical RL and multi-component reward that considers both health and socioeconomic cost. SiRL (Bushaj et al., 2023) uses a detailed ABM with RL and feeds the compartmental levels to the agent and uses rewards that are tied to intervention effectiveness and disease spread. These approaches excel at macro-level policy optimization such as determining lockdown time, resource allocation, and intervention sequencing (Mai et al., 2023; Arango & Pelov, 2020; Khadilkar et al., 2020). While valuable for analyzing epidemic policy, these centralized approaches do not examine the impact of different reward formulations on the learned RL policy.

**Addressing the Reward Engineering Gap**. This work addresses this gap by providing a systematic framework for evaluating reward formulation in spatial epidemic simulations. Instead of assuming particular behavioral patterns, we investigate the impact of different reward structure, varying environmental conditions such as partial observability, different infection rates, population density, distance decay factor (strength of disease transmission over distance), effectiveness of adherence to non-pharmaceutical interventions and spatial constraints, on learned behavior.

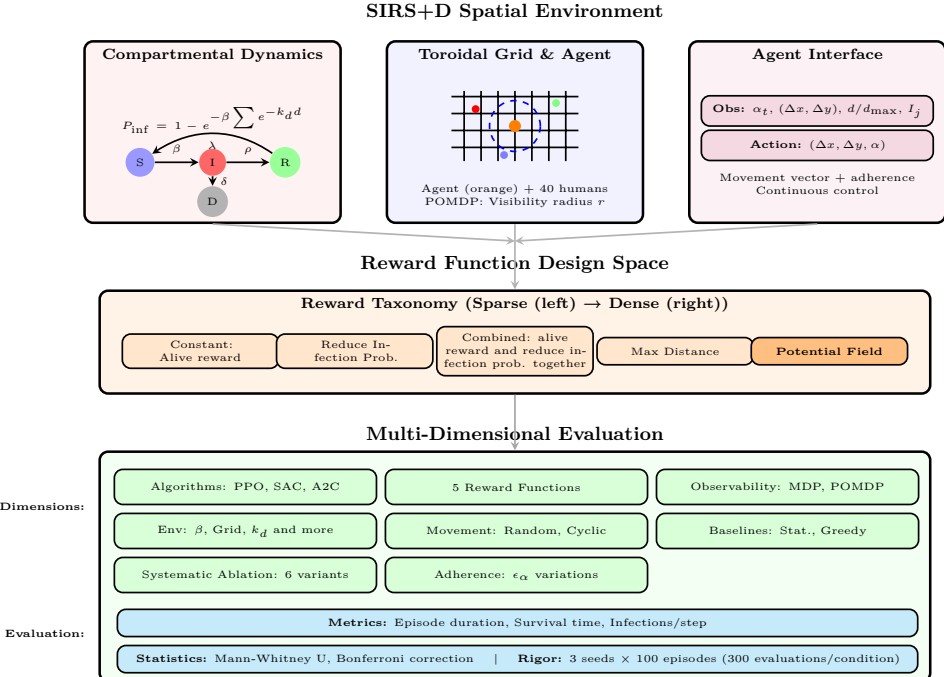

Figure 1: **ContagionRL System Architecture. Top:** SIRS+D spatial epidemic environment with toroidal grid, configurable observability, and continuous agent control interface. **Middle:** Reward function design from sparse to potential field-based rewards. **Bottom:** Multi-dimensional experimental evaluation.

## 3 Methodology

CONTAGIONRL is a custom Gymnasium-compatible (Towers et al., 2024) environment that integrates a SIRS+D compartmental epidemiological model (Hethcote, 2000). Built within an agent-based modeling framework, the environment simulates a single reinforcement learning (Sutton & Barto, 2018) agent interacting with a population of humans whose behaviors may be stochastic or deterministic or mix of both. This design enables the study of learned risk-averse navigation and adherence behaviors in response to an evolving epidemic (Saad-Roy & Traulsen, 2023) in different contexts.

**Single Agent Design Rationale**. Epidemic dynamics are inherently multi-agent interactions (Manfredi & D'Onofrio, 2013; Wang et al., 2016). However, our single-agent approach is a methodological choice that enables transparent attribution of learned behaviors to reward function design, which is the core scientific question of this work. Multi-agent environments introduce complexities such as non-stationarity due to other learning agents (Foerster et al., 2017), emergent social behavior (Leibo et al., 2017) and strategic interactions (Tampuu et al., 2017). These factors obscure whether observed behaviors stem from game-theoretic equilibria or from the reward structure itself. Single-agent formulation permits the identification of essential reward components for effective behavior learning. Understanding individual-level reward-behavior relationships is a necessary prerequisite for designing effective reward systems in multi-agent deployments where each agent still learns from its local reward signal. By isolating a single learning agent in a controlled population of non-learning humans, we can conduct detailed assessments of how different reward formulations affect the learned strategies without interference from adaptive social behaviors. The controlled nature of this methodology allows insights into the relationship between reward and behavior, that will be transferable to more complex multi-agent deployments where understanding individual-level incentive responses becomes critical for designing effective reward systems across a population.

### 3.1 SIRS+D Simulation Environment

**Environment Dynamics**. The simulation happens over discrete timesteps within a two-dimensional grid space of size $G \times G$ with periodic boundary conditions (Allen et al., 2009; Keeling & Rohani, 2008; Frenkel & Smit, 2002). The environment is populated by $N$ non-learning humans and a single controllable Reinforcement Learning (RL) agent. Each individual, human or agent, can exist in one of four states: Susceptible (S), Infected (I), Recovered (R), or Dead (D). We conduct a set of sensitivity analysis in Appendix H and vary the infection rate of disease ($\beta$) in Figure 12 and Table 15, vary the grid size and population density in Figure 13 and Table 16, vary the adherence effectiveness ($\alpha_{\text{eff}}$) in Figure 14 and Table 17, and vary the distance decay of disease transmission in Figure 15 and Table 18.

**Human Movement and State Transitions**. Non-learning humans exhibit stochastic or deterministic movement based on a specified movement type with a defined movement scale. Stochastic humans make their decisions by sampling from a Gaussian distribution within the bounds of the action space used to mimic the noisy nature of observation perception by the learning RL agent. Conversely, the deterministic non-learning humans are used to mimic the social and spatial constraints of a pandemic such as commuting. Their state transitions are governed by probabilistic rules. A susceptible human $h_s$ becomes infected with a probability $P_{\text{inf}}(h_s)$, determined by their exposure to nearby infected individuals $h_i$. Exposure is calculated based on the distance $d(h_s, h_i)$ between the susceptible human and each infected human, potentially limited by a maximum distance. The infection probability follows $P_{\text{inf}}(h_s) = 1 - \exp(-\beta \sum_{h_i} e^{-k_d \cdot d(h_s, h_i)})$, where $\beta$ is the base infection rate and $k_d$ is the distance decay factor. Once infected, a human recovers with a fixed probability $\rho$ at each timestep. After recovery, a human may lose immunity and revert to the susceptible state with a probability $\lambda$ at each timestep. Infected humans may also die with a probability $\delta$ at each timestep. Dead humans are removed from contributing to infection spread and cease movement. Appendix C.2 contains a description of environmental parameters and their descriptions.

**Agent Dynamics and Control**. The agent operates under SIRS+D dynamics identical to other individuals but differs in that it actively selects its movement and adherence levels via a learned policy. These choices affect both its spatial trajectory and susceptibility to infection. For a detailed formalization of the agent's action space, including adherence effects and boundary handling, see Section 3.2. Figure 8 shows a render of the environment.

**Initialization and Episode Structure**. Each simulation run begins with the agent positioned on the grid's center. The non-agent humans are distributed randomly, with an parameterization number of individuals set to the infectious (I) state. These initial infectees are placed at least a minimum safe distance from each other and from the agent. All remaining humans commence in the 'S' state, ensuring a minimum initial distance from the agent's starting position. The minimum safe distances are used to prevent trivial early termination of a simulation. The agent itself always starts as Susceptible ('S') with a predefined initial adherence level. To counteract stochastic extinction of the disease and trivial success episodes, a configurable reinfection mechanism is employed. If the number of infected individuals reaches zero, a specified number of susceptible humans (located beyond the minimum safe distance from the agent) are randomly chosen and transitioned back to the 'I' state. An episode concludes upon reaching the maximum simulation time limit. Termination can occur early if the agent transitions to infected ('I' state), or if the infection is eradicated and the reinfection mechanism cannot be activated due to insufficient eligible susceptible individuals.

### 3.2 RL Agent Formulation

We implement the agent's learning problem as a configurable decision-making process that supports both Markov Decision Process (MDP) and Partially Observable MDP (POMDP) formulations. We use both the MDP and POMDP formulation in our experiments. The POMDP employs a threshold-based visibility constraint, where agents can only observe other entities within a specified Euclidean distance radius. The agent's policy $\pi(a_t|o_t)$, mapping observations to actions, is learned using three deep reinforcement learning algorithms: Proximal Policy Optimization (PPO) (Schulman et al., 2017), Soft Actor-Critic (SAC) (Haarnoja et al., 2018), and Advantage Actor Critic (A2C) a synchronous variant of A3C algorithm (Mnih et al., 2016), all implemented via the STABLE-BASELINES3 library (Raffin et al., 2021).

**Observation Space**. The observation space provides the agent with information about its internal state and the surrounding environment at each timestep $t$. The observation $o_t$, constructed via the environment's internal observation method, is comprising the agent's current adherence level $\alpha_t$ (a scalar in $[0,1]$) and a flattened feature vector describing the non-learning population. This vector aggregates information for each non-learning human $h_j$. For every human $h_j$, the features consist of the relative position $(\Delta x_{aj}, \Delta y_{aj})$, representing the normalized displacement from the agent calculated considering toroidal boundaries and scaled by grid size (values in $[-0.5, 0.5]$); the normalized distance $(d_{aj}/d_{\max})$, which is the Euclidean distance scaled by the maximum possible grid diagonal distance $d_{\max}$; and a binary infection status indicator $(I_j)$.

**Action Space**. At each timestep $t$, the agent selects a continuous-valued action $a_t = (\Delta x, \Delta y, \alpha)$, where $(\Delta x, \Delta y) \in [-1, 1]^2$ defines a movement vector and $\alpha \in [0, 1]$ specifies the adherence level to non-pharmaceutical interventions (NPIs). NPIs are methods used to reduce the spread of an epidemic disease without any pharmaceutical drug treatments such as wearing masks or social distancing and are considered extremely effective ways of controlling primary outbreaks that reduce disease transmission and consequentially mortality (Rizvi et al., 2021; Ferguson et al., 2020). The agent's position is updated using the movement vector, scaled by a maximum step size, and wrapped around the grid's periodic boundaries:

$$x_{t+1} = (x_t + \Delta x) \bmod G, \quad y_{t+1} = (y_t + \Delta y) \bmod G, \tag{1}$$

where $G$ is the grid size. The adherence level $\alpha$ directly influences the agent's probability of infection upon exposure to nearby infected individuals by modulating the effective infection rate:

$$\beta_{\text{eff}} = \beta \cdot (\epsilon_\alpha + (1 - \epsilon_\alpha)(1 - \alpha)) \tag{2}$$

where $\epsilon_\alpha \in (0, 1)$ represents the residual infection risk at full adherence. Thus, higher adherence values reduce susceptibility, but never eliminate risk entirely.

### 3.3 Reward Functions

To investigate the impact of reward engineering on policy learning within the CONTAGIONRL environment, we implemented and evaluated several distinct reward functions. These functions vary in complexity and the specific agent behaviors they aim to incentivize, with the overarching goal of prolonging the agent's susceptible state. The simplest among these is a Constant Reward, which provides a fixed positive signal if the agent remains uninfected. The Reduce Infection Probability reward directly incentivizes the agent to minimize its calculated likelihood of infection at each step, based on exposure to nearby infected individuals and its current adherence level. This can be employed as a Combined Reward by supplementing it with the constant bonus. Another approach, Maximize Nearest Distance, focuses on spatial dynamics, rewarding the agent for maintaining a distance from the closest susceptible or infected individuals, particularly beyond a predefined threshold.

The most complex of the explored designs is the Potential Field Reward. This function models interactions as forces, where other agents (particularly infected ones) exert repulsive forces on the agent. The reward encourages movement that aligns with the net resultant force vector, effectively guiding the agent away from high-risk areas. This directional guidance is complemented by terms rewarding the agent for its health status and NPI adherence. These reward function are tested in different environmental configuration. Details of the reward calculation are provided in Appendix C.3.

An important design consideration is the information scope of each reward component. The health and adherence terms depend solely on the agent's own state and action, making them purely local signals. In contrast, the directional and magnitude components of the potential field reward are computed using the positions and states of *all* humans in the simulation, regardless of the agent's visibility radius. This creates a deliberate information asymmetry between the agent's observations and the reward signal: under POMDP conditions, observations respect the visibility constraint (Section 3.2), while the reward provides a globally informed training signal. This design functions as a form of privileged critic (Pinto et al., 2017), where the reward shapes policy gradients using complete state information during training, but the learned policy must generalize from local observations alone at evaluation time. Table 6 summarizes the information scope of each reward component.

## 4 Baselines

To contextualize the performance of the learned policies with RL, we established several non-learning deterministic baselines. These policies represent simple, fixed strategies and serve as benchmarks to evaluate the complexity and effectiveness of the learned behavior. We implemented three distinct baseline strategies (see Appendix C.6 for more details).

**Stationary Baseline**. This agent remains immobile with zero adherence to NPIs ($\Delta x = \Delta y = \alpha = 0$) throughout the episode. Its performance depends entirely on initial conditions and the stochastic behavior of the surrounding population.

**Random Baseline**. At each timestep, this agent samples movements $\Delta x, \Delta y \sim \mathcal{U}(-1, 1)$ and adherence $\alpha \sim \mathcal{U}(0, 1)$, while being oblivious with regard to the environment and observations entirely. It serves as a reference for performance under uninformed, mobile behavior.

**Greedy Distance Maximizer Baseline**. This heuristic policy implements a specific survival strategy focused on immediate risk avoidance. At each step, it first identifies the nearest infected human. It then evaluates a predefined set of potential movement actions across the eight cardinal and intercardinal directions, plus staying stationary. It systematically simulates the outcome of each potential action, by computing the hypothetical distance to the nearest infected human if that move were taken, and selects the action that yields the greatest distance separation. The policy's evaluation of counterfactual outcomes provides a situational awareness absent in simpler baselines. While myopic and handcrafted, this policy provides a strong heuristic benchmark by explicitly evaluating the spatial consequence of each potential action.

## 5 Results

**Performance of different algorithms in ContagionRL environment**. To evaluate the utility of the CONTAGIONRL for obtaining effective learned policies via reinforcement learning (RL), we trained and assessed three distinct RL algorithms: Soft Actor-Critic (SAC) (Schulman et al., 2017), Proximal Policy Optimization (PPO) (Schulman et al., 2017), and Advantage Actor-Critic (A2C) (Mnih et al., 2016). Their performance was benchmarked against baselines mentioned in section 4 (configuration details are in Appendix C.2 and Appendix C.5).

Figure 2 shows the distribution of episode durations for both baselines and the RL agents. The distributions of episode duration for the stationary and random baselines have substantially less variance compared to those of the RL agents and the greedy heuristic. To determine the statistical significance of these different distributions, we conducted pairwise comparisons using the Mann-Whitney U test. The outcomes of these tests, including both two-sided and one-sided $p$-values, are presented in Table 1.

The statistical analyses reveal that the learned policies derived from the RL approaches (PPO, SAC, and A2C) significantly outperform both the random and stationary baselines. The greedy heuristic baseline demonstrates superior performance in pairwise comparisons against all three RL agents, achieving statistically significantly longer episode durations (Table 1). However, as depicted by the mean episode durations and their 95% confidence intervals in Figure 9, the performance of the RL agents is comparable to that of the greedy baseline, with overlapping confidence intervals.

This tells us that while the distributions of episode durations and their medians, differ significantly in favor of the greedy approach, the mean performance of greedy does not outperform learned RL policies. These results collectively demonstrate several key points: (1) The observation formulation within the CONTAGIONRL environment provides a sufficient and informative signal for the RL agents to develop effective control policies that substantially exceed naive baselines. (2) The environment supports the application and comparative evaluation of various RL algorithms, highlighting their compatibility with its inherent stochasticity and complex dynamics, leading to policies competitive with a strong heuristic. (3) While a specialized greedy heuristic can achieve statistically superior median performance, the learned RL agents develop sophisticated strategies that yield comparable mean performance, showing the environment's utility for investigating adaptive decision-making processes that can approach the efficacy of well-designed heuristics without explicit rule-based programming used in ABMs.

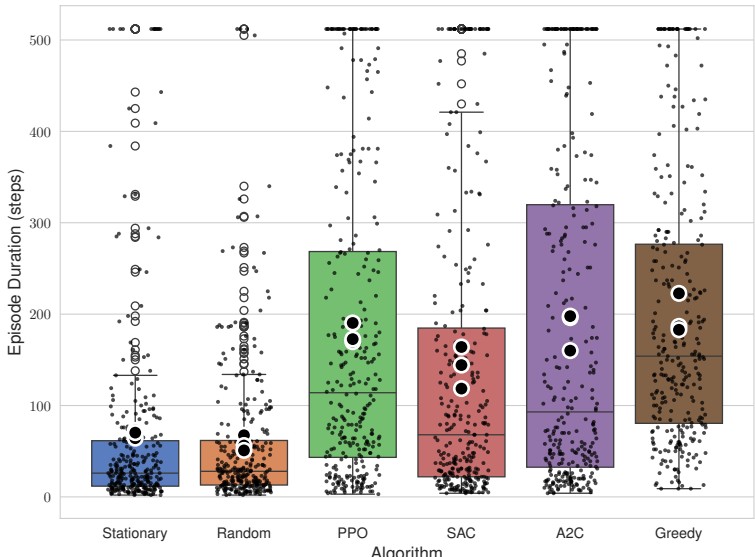

Figure 2: Episode duration distributions across different agents, including learning-based (PPO, SAC, A2C) and non-learning baselines (Random, Stationary, Greedy). Each small black dot represents one episode in the totality of episodes across 3 seeds × 100 evaluation runs. Per-seed means are shown as large black dots with white outlines. This figure complements the summary statistics in Figure 9 and statistical comparisons in Table 1.

Table 1: Pairwise comparisons of episode durations between agents using the Mann–Whitney U test. Two-sided $p$-values assess distributional differences, and one-sided $p$-values (Bonferroni-corrected) test for performance advantage. The **Winner** is the agent with significantly longer episode duration (after correction); "–" indicates no significant difference. Significance codes: * $p < 0.05$, ** $p < 0.01$, *** $p < 0.001$, n.s. = not significant.

| Agent A | Agent B | $p$ (2-sided) | $p$ (1-sided) | Sig (2) | Corrected $p$ (1) | Sig (1) | Winner |
|---|---|---|---|---|---|---|---|
| Stationary | Random | 0.5594 | 0.7204 | n.s. | 1 | n.s. | – |
| Stationary | PPO | 3.22e-25 | 1.61e-25 | *** | 2.42e-24 | *** | PPO |
| Stationary | SAC | 3.83e-13 | 1.92e-13 | *** | 2.88e-12 | *** | SAC |
| Stationary | A2C | 2.46e-22 | 1.23e-22 | *** | 1.84e-21 | *** | A2C |
| Stationary | Greedy | 3.87e-48 | 1.94e-48 | *** | 2.91e-47 | *** | Greedy |
| Random | PPO | 1.39e-25 | 6.96e-26 | *** | 1.04e-24 | *** | PPO |
| Random | SAC | 1.27e-12 | 6.32e-13 | *** | 9.49e-12 | *** | SAC |
| Random | A2C | 1.29e-22 | 6.45e-23 | *** | 9.68e-22 | *** | A2C |
| Random | Greedy | 9.96e-51 | 4.98e-51 | *** | 7.47e-50 | *** | Greedy |
| PPO | SAC | 0.00103 | 0.00052 | ** | 0.00775 | ** | PPO |
| PPO | A2C | 0.7604 | 0.62 | n.s. | 1 | n.s. | – |
| PPO | Greedy | 0.000639 | 0.000320 | *** | 0.00479 | ** | Greedy |
| SAC | A2C | 0.00430 | 0.00215 | ** | 0.0323 | * | A2C |
| SAC | Greedy | 1.06e-12 | 5.28e-13 | *** | 7.93e-12 | *** | Greedy |
| A2C | Greedy | 0.000341 | 0.000171 | *** | 0.00256 | ** | Greedy |

**Comparison of Different Reward functions in ContagionRL**. Our primary algorithm evaluations utilized a potential field-based reward. Now, we investigate the differential impact of diverse reward structures on performance. We use the environment configuration in Appendix C.2 and alter the reward functions to gauge difference in performance with PPO. The principal objective for the agent remains the avoidance of infection. This is quantified by the episode duration, which measures the number of timesteps during which the agent remains susceptible before experiencing its first infection. In other words, if the agent becomes infected for the first time at timestep $t$, the achieved episode duration is $t - 1$. Although alternative reward

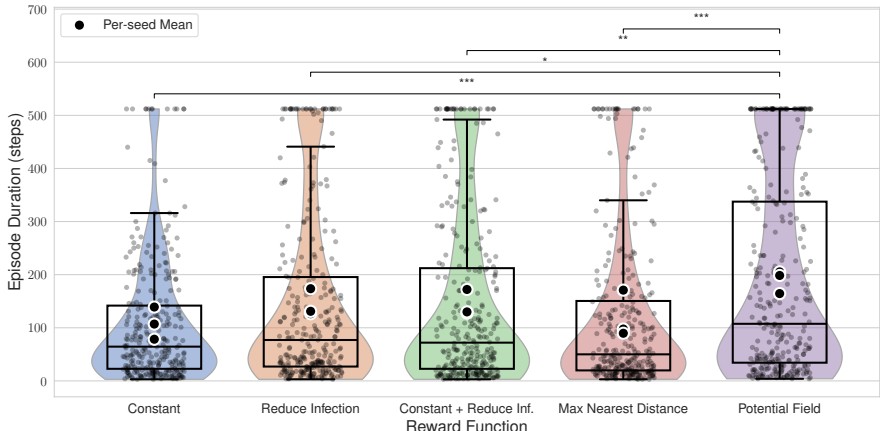

Figure 3: Comparison of PPO agent performance under five different reward functions. Each model was trained with three random seeds and evaluated over 100 episodes per seed (300 episodes total per reward function). Violin plots show the distribution of episode durations, overlaid with boxplots and per-episode results (black points). Large black dots with white outlines indicate the per-seed mean. One-sided Mann–Whitney U tests (with Bonferroni correction) compare each reward function to the *Potential Field* baseline. Statistically significant differences (* $p < 0.05$, ** $p < 0.01$, *** $p < 0.001$) are annotated. See Table 10 for exact test values and Figure 10 for corresponding means with confidence intervals.

schemes could be designed to elicit and study different behaviors, our focus here is on optimizing for this primary goal.

Figure 3 and Figure 10 visually summarize the performance distributions and mean episode durations, respectively, for policies trained under five distinct reward functions (see Section 3.3 and Appendix C.3). The statistical significance of these differences is further detailed in Table 10, which presents pairwise Mann-Whitney U test comparisons against the Potential Field baseline.

The results indicate that the Potential Field reward function, via its dense and spatially informed feedback, results in a significantly superior performance, achieving the longest mean episode durations. Policies trained with the Constant reward structure show poor performance. We attribute this to the sparse and uninformative nature of a constant bonus, which provides little guidance for nuanced decision-making. Similarly, the Max Nearest Distance reward function also resulted in significantly shorter episode durations compared to the Potential Field approach.

Both the Max Nearest Distance and the Reduce Infection probability reward functions may guide the agent towards suboptimal local optimas. An agent might learn to maintain a specific distance from currently infected individuals, thereby maximizing its immediate reward. However, it fails to adopt a globally optimal positioning strategy that minimizes long-term infection risk. This highlights the challenge of designing reward functions that encourage far-sighted behavior rather than myopic optimization in complex spatial settings.

An important observation, across all tested reward functions, is the consistent emergence of high NPI adherence in the learned policies. This suggests that the benefit of maximizing adherence is a relatively identifiable signal within the CONTAGIONRL environment, discoverable through learning irrespective of the specific reward structure guiding navigation. A detailed analysis of learned adherence behavior across all reward functions and ablations is provided (Appendix E). The primary challenge lies in learning effective spatial navigation strategies, a task for which the choice of reward function is more important. The data presented in Figures 3 and 10, and substantiated by the statistical comparisons in Table 10, unequivocally demonstrate the influence of reward function design on the learned policy to achieve the desired infection avoidance behavior.

**Population-Level Epidemic Impact.** To assess whether the agent's learned behavior affects population-wide dynamics, we measure the infection rate: mean new S→I transitions per timestep among non-agent

Table 2: Population-level epidemic impact of learned policies. **New Inf./step** reports the mean number of new S→I transitions per timestep among the 40 non-agent humans, averaged over 90 episodes (3 seeds × 30). Lower values indicate reduced epidemic burden. $\Delta\%$ columns show the percentage reduction relative to each baseline. One-sided Mann–Whitney U tests with Bonferroni correction (5 comparisons per baseline). Significance: * $p < 0.05$, ** $p < 0.01$, n.s. = not significant.

| | | vs Random | | vs Static | |
| --- | --- | --- | --- | --- | --- |
| **Policy** | **New Inf./step** | $\Delta\%$ | **Sig.** | $\Delta\%$ | **Sig.** |
| Random (baseline) | 1.133 | — | — | — | — |
| Static (baseline) | 1.121 | — | — | — | — |
| Potential Field | 0.890 | $-21.4\%$ | ** | $-20.6\%$ | ** |
| Const.+Red. Inf. | 0.925 | $-18.3\%$ | ** | $-17.5\%$ | ** |
| Constant | 0.963 | $-15.0\%$ | n.s. | $-14.2\%$ | * |
| Max Near. Dist. | 0.972 | $-14.2\%$ | n.s. | $-13.3\%$ | n.s. |
| Reduce Infection | 1.007 | $-11.1\%$ | n.s. | $-10.2\%$ | n.s. |

humans, which controls for differences in episode length across policies. Table 2 shows that trained policies reduce population-level infection rates by 10–21% relative to baselines, with the Potential Field reward achieving the largest reduction ($p < 0.01$, Bonferroni-corrected). This ranking is broadly consistent with individual survival performance (Figure 3), consistent with the mechanism that agents avoiding infection remove themselves as transmission vectors. Given that the agent constitutes only one of many entities, this represents a conservative lower bound on the impact of learned avoidance behavior.

**Ablations of reward function**. To understand the contributions of individual components to the overall efficacy of the potential field reward function, we conduct a systematic ablation study. Specific elements of the full potential field (FPF) reward were individually removed, and the performance of policies trained with these ablated reward structures was compared against the FPF baseline. We use the same configuration as the previous section but changed the value of the reward ablation field (Appendix C.2). We did the following ablations: removal of the movement magnitude component (No Magnitude), removal of the directional component of movement (No Direction), removal of the entire movement-related reward (No Movement), removal of the adherence-based reward (No Adherence), removal of the health status reward (No Health), and removal of the repulsion force from susceptible individuals (No Susceptible Repulsion). The comparative performance distributions are illustrated in Figure 4, mean performances are shown in Figure 11, and detailed statistical comparisons are presented in Table 11.

The results (Table 11) indicate that several components are important to the FPF reward's success. Specifically, the removal of the directional guidance (No Direction), the overall movement incentive (No Movement), or the reward for NPI adherence (No Adherence) led to a statistically significant and substantial degradation in performance of learned policy compared to the FPF as our baseline. These elements provide the primary signal guiding the agent's navigation to safer regions. Without this explicit directional information, the agent's ability to learn effective spatial strategies is severely impaired.

The ablation of the movement magnitude component (No Magnitude) did not result in a statistically significant performance change, suggesting that the direction of the suggested movement is more important than its precise scaling in this context. Similarly, removing the basic health reward (No Health) or the repulsion from susceptible individuals (No Susceptible Repulsion) did not significantly impair performance, implying these components are either less influential or their effects are implicitly captured by other elements of the FPF reward.

Upon removing the adherence reward component (No Adherence), we observed a significant degradation in agent performance. While we might expect agents to implicitly learn the importance of high NPI adherence due to its direct impact on infection risk (especially under simpler reward schemes), this was not what we observed with the complex FPF reward. It is evident from our findings that when a diverse set of reward signals is provided, explicitly rewarding critical behaviors like NPI adherence is crucial for effective learning,

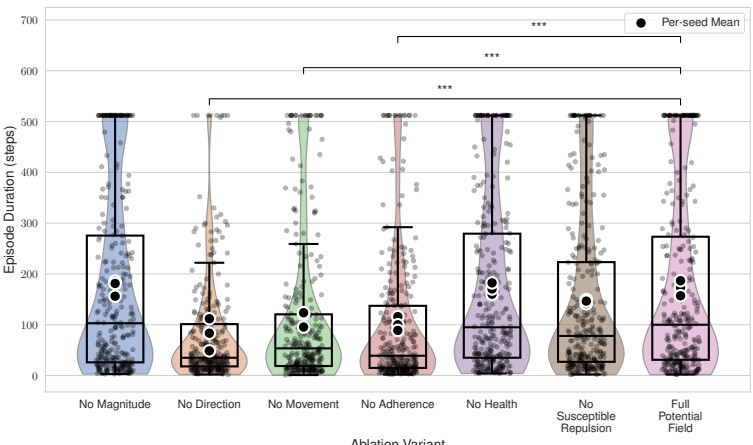

Figure 4: Ablation study of the *Potential Field* reward function. Each variant was evaluated over 100 episodes across 3 training seeds (300 episodes total). Violin plots show the distribution of episode durations, overlaid with boxplots and individual episode results (small black dots). Large black dots with white outlines represent per-seed means. One-sided Mann–Whitney U tests (Bonferroni-corrected) compare each ablation to the full model (*Full Potential Field*), with significance annotations (* $p < 0.05$, ** $p < 0.01$, *** $p < 0.001$) shown for statistically significant differences. See Figure 11 for aggregated means with confidence intervals, and Table 11 for full statistical test results.

as opposed to depending on the policy to deduce their significance. This highlights the importance of careful component weighting and inclusion when engineering sophisticated reward functions for complex tasks.

**Comparison of Partial Observability (POMDP) and Full Observability**. To understand the learned behavior under limited visibility, we formulate the epidemic control problem as a partially observable Markov decision process (POMDP). In real epidemics, actors have incomplete information about the environment state and infection status of individuals (Gersovitz & Hammer, 2004; Geoffard & Philipson, 1996; Farboodi et al., 2021). We implement this POMDP formulation using a visibility radius constraint, where individuals within radius $r$ of RL agent's position are passed as observations and individuals beyond this radius are unobservable. This natural information asymmetry captures a realistic scenario where agents have limited sensing or information networks that decay with distance. Figure 5 demonstrates trained partial visibility models (r=10, r=15, r=20) paradoxically outperform the baselines and the full visibility trained model, Trained Full, in average reward, episode length and infections per timestep. This counter-intuitive result suggests that complete information can be detrimental to learning, potentially due to observation noise and the increased dimensionality in high-dimensional observation spaces (see Appendix F for feature importance analysis). Limited observability forces the agent to develop more robust strategies and reduces the complexity of the decision space. The trained (Full) model's performance is on par with the greedy heuristic. Notably, a performance differences between r=10, r=15 and r=20 conditions are not statistically significant according to Table 12, indicating a performance plateau beyond a certain visibility threshold. We note that the potential field reward is computed from global state even under POMDP conditions (Section 3.3), functioning as a privileged training signal. The agent's improved performance under limited visibility therefore reflects genuine policy robustness rather than information leakage through observations.

The POMDP perspective reveals that effective epidemic control strategies can come from local information processing. Agents are able to infer general patterns from partial observations. The improved performance occurring under constrained visibility shows that an information bottleneck can enhance robustness.

**Social and Economic Constraint in Human Movement**. Mobility is a key consideration in epidemiological models (Fakir & Bharati, 2021) and literature review shows that mobility patterns remain unchanged in a pandemic for essential or low wage workers (Gallagher et al., 2021). Inspired by this, we conducted an experiment to evaluate the performance of the agent under these socioeconomic constraints in mobility. Figure 6 shows that the trained PPO agent outperforms stationary, random and greedy baselines when trained on a

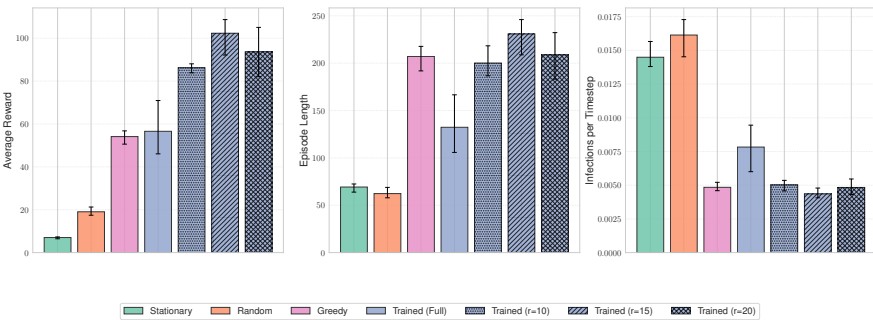

Figure 5: Impact of visibility radius constraints on RL agent performance in epidemic control. The figure compares four agent types across different observation capabilities: Full Visibility (agent observes all infected individuals), and Limited Visibility with radius constraints r=10, r=15, and r=20 (agent only observes infected individuals within the specified radius). Agent types include: Stationary (no movement), Random (random actions), Greedy (heuristic policy avoiding nearest infected), and Trained (PPO-trained RL agents with respective visibility constraints). Results averaged across 3 random seeds with 100 evaluation episodes per seed (N=300 per condition). Error bars show 95% bootstrap confidence intervals from per-seed means. Left: Average cumulative reward per episode. Center: Episode length (survival time in timesteps). Right: Infections per timestep calculated from total infections divided by episode length. Hatching patterns distinguish trained variants: dots (r=10), diagonal lines (r=15), and crosses (r=20). Limited visibility agents (r=15, r=20) achieve higher performance than full visibility, suggesting that observation constraints can improve learning by reducing observation noise and focusing attention on nearby threats.

workplace/home movement cycle. We notice that the stationary and random baselines declined in this cyclic movement, but the trained and greedy baseline improve considerably. Furthermore, the trained RL agent has a higher average reward and higher mean episode length, with an extremely low rate of infections per timestep (Table 3). In the random environment, the agent struggles with long predictions and has a short horizon on its reward, as it performs worse than the greedy baseline. We attribute its inability to beat the local optimization of the greedy algorithm to the noisy observations and movements of the random environment. By incorporating the spatio-temporal regularities of a workplace/home mobility cycle, a trained agent can formulate more effective long-term strategies. The resulting expansion of its planning horizon is empirically validated by a substantial improvement in cumulative average reward over an agent exposed to a random mobility pattern. We anticipated that trained agents operating under cyclic movement patterns would exhibit lower standard deviation across performance metrics compared to those under random movement. However, our results contradicted this expectation. Furthermore, while infections per timestep increased for all agent types, the trained agent demonstrated no such increase, maintaining stable infection rates throughout the evaluation period.

## 6  Conclusion

We introduced CONTAGIONRL, a novel reinforcement learning platform that enables systematic evaluation of reward function design and for analysis of learned behavior in a spatial epidemics model. Through comprehensive experiments across multiple RL algorithms (PPO, SAC, A2C) and diverse environmental conditions, we show that reward engineering dramatically influences the learned survival strategies. Systematic reward ablations reveal that directional guidance and adherence incentives are essential components for robust policy learning while agents consistently learn adherence to non-pharmaceutical interventions regardless of the reward formulation used for spatial navigation.

We show that different RL algorithms can successfully learn mitigation strategies in the environment. We compare different reward function formulations on performance and structural ablation of reward function design. Also, we find that constraining agent visibility yields superior performance relative to full observability, attributable to reduced dimensional complexity. Furthermore, we reveal structured movement compared to random movement leads to increased longevity of the trained agent. The CONTAGIONRL environment

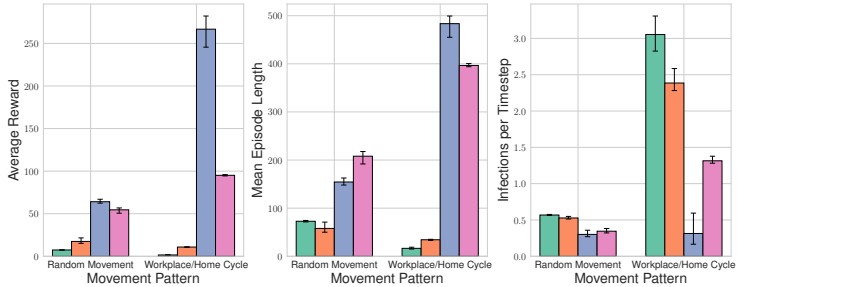

Figure 6: Performance comparison of RL agents across different human movement patterns in epidemic control. The figure shows three metrics across two movement patterns: Random Movement (continuous random walk) and Workplace/Home Cycle (structured movement between workplace and home locations). Four agent types are compared: Stationary, Random, Greedy, and Trained (PPO agent). Results are averaged across 3 random seeds with 100 evaluation episodes per seed. Error bars show 95% bootstrap confidence intervals from per-seed means. Left: Average cumulative reward per episode. Center: Mean episode length (survival time). Right: Infection rate. Trained agents significantly outperform all baselines in the Workplace/Home Cycle environment but show mixed results in Random Movement. Statistical results can be found in Table 3

Table 3: Statistical significance tests comparing RL agent performance across movement patterns shown in Figure 6. Trained agents were evaluated using Mann-Whitney U tests with Bonferroni correction for multiple comparisons ($\alpha = 0.05$). Each condition used 300 episodes (3 seeds × 100 episodes per seed). Cross-condition compares trained agents between movement patterns.

| Movement Pattern | Comparison | Winner | U-stat. | p-value | Sig. |
|---|---|---|---|---|---|
| Random Movement | Trained vs Stationary | Trained | 80,724.0 | $p < 0.001$ | *** |
| | Trained vs Random | Trained | 69,720.0 | $p < 0.001$ | *** |
| | Trained vs Greedy | Trained | 49,117.0 | $p = 0.420$ | n.s. |
| Workplace/Home Cycle | Trained vs Stationary | Trained | 89,067.0 | $p < 0.001$ | *** |
| | Trained vs Random | Trained | 87,819.0 | $p < 0.001$ | *** |
| | Trained vs Greedy | Trained | 85,853.0 | $p < 0.001$ | *** |
| Cross-Condition | Trained Random vs Trained Workplace | Workplace | 4,100.0 | $p < 0.001$ | *** |

offers insights for developing behaviorally informed epidemic intervention protocols while enabling systematic evaluation of reward function design. It also provides a form of advancing our understanding of how reward structure shapes emergent behaviors in complex spatial and temporal systems such as spatial epidemic models.

**Broader Impact Statement**. ContagionRL provides a platform for researchers to study how reward design influences learned behaviour in an epidemic setting. In a broad sense, experiments performed using platforms of this type can help us understand how potential public health intervention strategies, leading to perceived rewards or avoiding penalties, might shape protective behaviours. Nonetheless, we emphasize that human behaviour around disease avoidance is complex and heterogeneous, and that the relationship between simulation-encoded rewards and real-world perceived rewards is also challenging to understand. Simulation findings will complement rather than replace epidemiological expertise and empirical validation for the foreseeable future.

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

## Appendix

# A    Compartmental Epidemiology Models Continued

## A.1    Notation and parameter definitions

We denote by the numbers of susceptible, infected, recovered, exposed and deceased individuals at time $t$ using $S(t)$, $I(t)$, $R(t)$, $E(t)$, $D(t)$, respectively. In the SIR, SIRS and SIRS+D models the total living population is

$$N(t) = S(t) + I(t) + R(t), \tag{3}$$

whereas in the SEIR model

$$N(t) = S(t) + E(t) + I(t) + R(t), \tag{4}$$

The model parameters are:

$\beta$  transmission rate (per-contact rate at which susceptibles become infected);

$\gamma$  recovery rate (rate at which infected individuals recover);

$\sigma$  progression rate (rate at which exposed become infectious, SEIR only);

$\xi$  immunity-waning rate (rate at which recovered lose immunity, SIRS and SIRS+D);

$\mu$  disease-induced mortality rate (rate at which infected die, SIRS+D only).

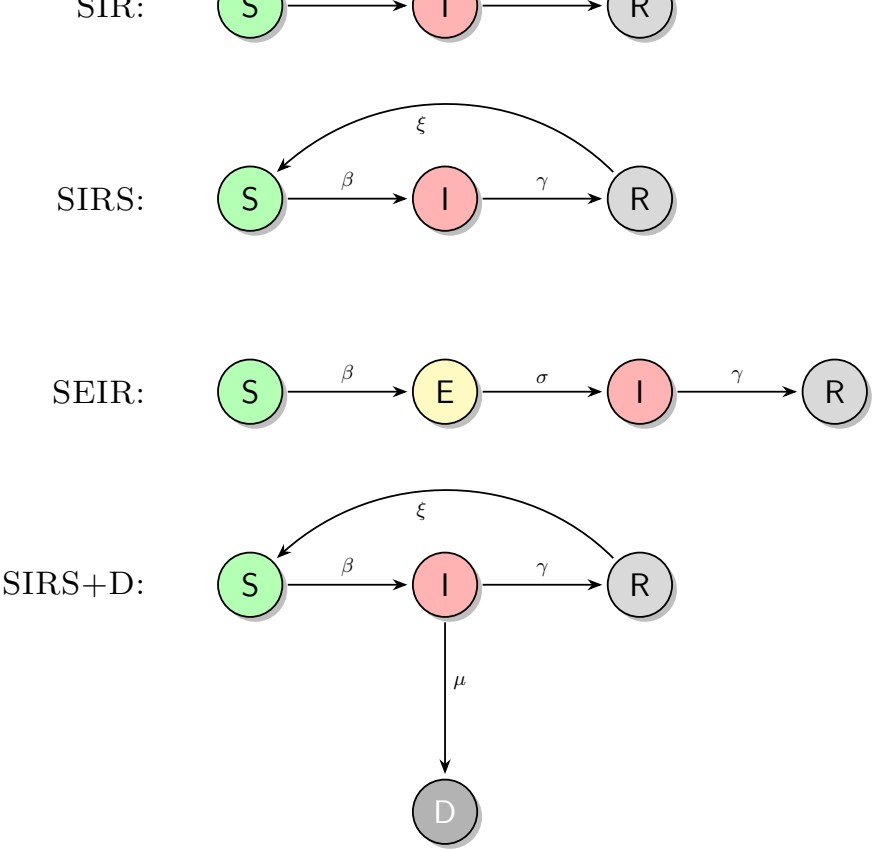

Figure 7: Compartmental epidemic models visualized: SIR, SIRS, SEIR, and SIRS+D.

## A.2 Model equations

The differential equations below provide an overview of how compartments are tracked and changed population-wide based on the defined respective parameters in Appendix A.1.

**SIR model.**

$$\frac{dS}{dt} = -\beta \frac{SI}{N}, \tag{5}$$

$$\frac{dI}{dt} = \beta \frac{SI}{N} - \gamma I, \tag{6}$$

$$\frac{dR}{dt} = \gamma I. \tag{7}$$

**SIRS model.**

$$\frac{dS}{dt} = -\beta \frac{SI}{N} + \xi R, \tag{8}$$

$$\frac{dI}{dt} = \beta \frac{SI}{N} - \gamma I, \tag{9}$$

$$\frac{dR}{dt} = \gamma I - \xi R. \tag{10}$$

**SEIR model.**

$$\frac{dS}{dt} = -\beta \frac{SI}{N}, \tag{11}$$

$$\frac{dE}{dt} = \beta \frac{SI}{N} - \sigma E, \tag{12}$$

$$\frac{dI}{dt} = \sigma E - \gamma I, \tag{13}$$

$$\frac{dR}{dt} = \gamma I. \tag{14}$$

**SIRS+D model.**

$$\frac{dS}{dt} = -\beta \frac{SI}{N} + \xi R, \tag{15}$$

$$\frac{dI}{dt} = \beta \frac{SI}{N} - (\gamma + \mu) I, \tag{16}$$

$$\frac{dR}{dt} = \gamma I - \xi R, \tag{17}$$

$$\frac{dD}{dt} = \mu I. \tag{18}$$

## A.3 Comparison of the epidemiological models

Table 4: Comparison of epidemic models and their capacity for individual-level and adaptive behavior modeling

| Model | Compartments | Reinfection | Incubation | Death | Behavioral Modeling | Adaptive Behavior Modeling |
|---|---|---|---|---|---|---|
| SIR (Kermack & McKendrick, 1927) | S, I, R | No | No | No | None | No |
| SIRS (Kermack & McKendrick, 1927) | S, I, R | Yes | No | No | None | No |
| SEIR (Kermack & McKendrick, 1927; Wilson & Worcester, 1945) | S, E, I, R | No | Yes | No | None | No |
| SIRS+D (Wilson & Worcester, 1945) | S, I, R, D | Yes | No | Yes | None | No |
| ABM (Grimm & Railsback, 2013) | Custom per agent | Yes | Optional | Yes | **Prescribed rules** | Partial |
| **ContagionRL** | **Modified SIRS+D** | **Yes** | **No** | **Yes** | **Learned policies (RL)** | **Yes** |

## B   Environment Visualization

Figure 8 shows a single frame of the environment's rendered video. The visualization provides a comprehensive and intuitive overview of the simulation state and the environment at each timestep. The panel at the center shows the environment's grid, in which each human is represented as a colored circle depending on its epidemiological state. The legend is shown on the left of the grid (blue for susceptible, red for infectious, green for recovered, and gray for dead). The focal agent, whose actions are being learned by RL, is shown as a orange dot with dark border. The small arrow, originating from the agent, shows its most recent movement direction and magnitude.

The upper panel on the right of the grid, shows the details related to the agent such as current state, position, adherence level, and cumulative reward. While the panel below it shows the population distribution across all epidemiological states in numbers and absolute percentages (due to increased need of processing power for rendering this function is turned off by default).

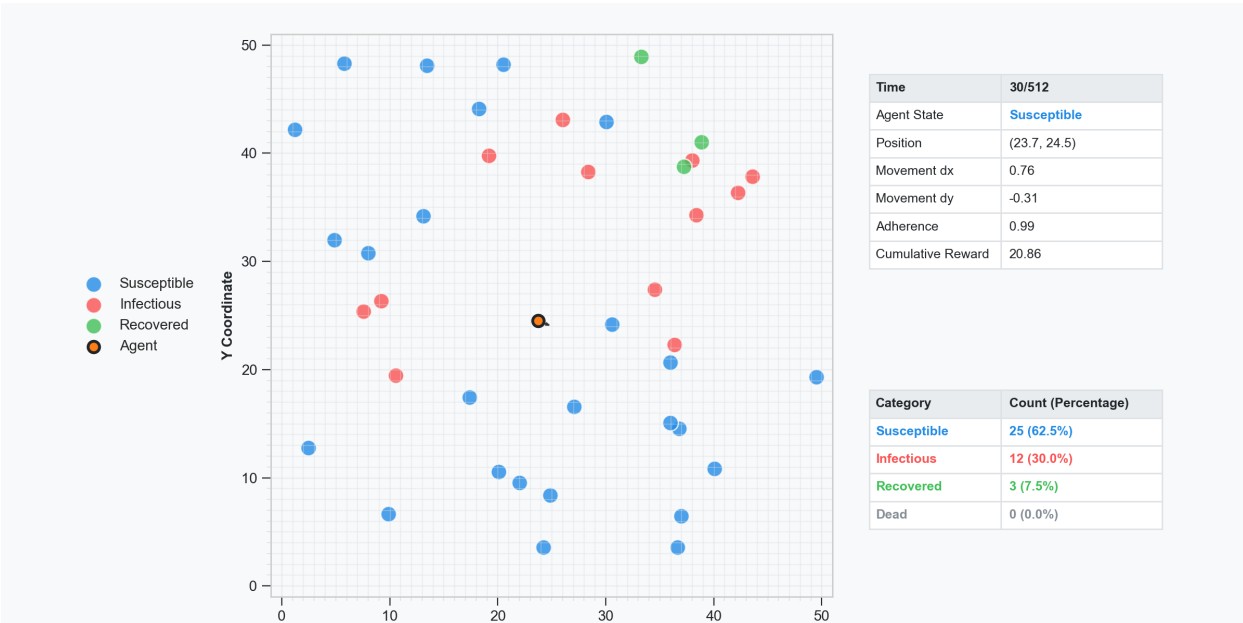

Figure 8: Sample render of the SIRS+D environment at step 30 of an episode

## C   Methodology Continued

### C.1   Toroidal Distance Calculation

Throughout the simulation, distances between individuals are calculated using the standard Euclidean distance metric, modified to account for the periodic boundary conditions of the toroidal grid. This ensures that the distance represents the shortest path, considering potential "wrap-around" across grid edges.

The distance $d(i, j)$ between two individuals $i$ and $j$ at positions $(x_i, y_i)$ and $(x_j, y_j)$ respectively, within a grid of side length $G$, is calculated as:

$$d(i, j) = \sqrt{\Delta x_{\text{wrap}}^2 + \Delta y_{\text{wrap}}^2} \tag{19}$$

where the wrapped differences along each axis are determined by:

$$\Delta x_{\text{wrap}} = \min(|x_i - x_j|, G - |x_i - x_j|), \Delta y_{\text{wrap}} = \min(|y_i - y_j|, G - |y_i - y_j|) \tag{20}$$

This formulation ensures that the distance along each dimension corresponds to the shorter of the two possible paths (direct or wrapped around the grid). This distance metric is fundamental to calculating infection exposure, determining the nearest threat for the Greedy policy, and computing forces in the potential field reward function.

## C.2 Environment Configuration Details

For completeness and to ensure the reproducibility of our experiments, this section provides a detailed breakdown of the specific parameter values used to configure the SIRS+D simulation environment described in section 3. Table 5 lists each parameter, its corresponding symbol used within the text (where applicable), its configured value, and a brief description of its role within the simulation dynamics. These parameters define the epidemiological characteristics, spatial properties, and agent interaction rules governing the environment used throughout our study.

Table 5: SIRS+D Environment Configuration Parameters

| Parameter | Value | Type | Range | Description |
|---|---|---|---|---|
| Simulation Time | 512 | integer | $[1, \infty)$ | Maximum number of timesteps per episode. |
| Grid Size | 50 | integer | $[1, \infty)$ | Side length of the square, toroidal grid (Value x Value). |
| Number of Humans | 40 | integer | $[0, \infty)$ | Number of non-agent individuals in the simulation. |
| Initial Infected | 10 | integer | $[0, 40]$ | Number of initially infected individuals (up to Number of Humans). |
| Infection Rate | 0.5 | float | $[0, 1]$ | Base infection transmission rate per exposure unit. |
| Initial Agent Adherence | 0 | float | $[0, 1]$ | Agent's NPI adherence level at the start of an episode. |
| Distance Decay | 0.3 | float | $[0, \infty)$ | Exponential decay factor for infection exposure based on distance. |
| Lethality Rate | 0 | float | $[0, 1]$ | Per-step probability of death for infected individuals. |
| Immunity Loss Probability | 0.25 | float | $[0, 1]$ | Per-step probability of transitioning from Recovered (R) to Susceptible (S). |
| Recovery Rate | 0.1 | float | $[0, 1]$ | Per-step probability of transitioning from Infected (I) to Recovered (R). |
| Adherence Penalty Factor | 1 | float | $[1, \infty)$ | Multiplicative factor for adherence cost (used by some reward functions). |
| Adherence Effectiveness | 0.2 | float | $[0, 1]$ | Residual transmission factor when agent adherence is 1.0 (e.g., 0.2 means 20% of beta remains). |
| Movement Type | `continuous_random` | string | Categorical | Movement model for non-agent individuals (e.g., `continuous_random`). |
| Movement Scale | 1 | float | $[0, 1]$ | Maximum step size scale for non-agent individuals. |
| Visibility Radius | -1 | float | $\{-1\} \cup [0, \infty)$ | Agent's observation radius (-1 indicates full grid visibility). |
| Reinfection Count | 5 | integer | $[0, \infty)$ | Number of S individuals reinfected if $I_t = 0$. |
| Safe Distance | 10 | float | $[0, \infty)$ | Min distance for infected S individuals relative to the agent. |
| Initial Agent Distance | 5 | float | $[0, \infty)$ | Min distance all individuals are placed from the agent at initialization. |
| Max Infection Distance | 10 | float | $\{-1\} \cup (0, \infty)$ | Max distance for an infected individual to contribute to exposure (-1 for infinite). |
| Reward Function Type | `potential_field` | string | Categorical | Identifier for the reward function (e.g., `potential_field`, `sparse`). |
| Reward Ablation | `full` | string | Categorical | Specifies variant of reward function for ablation studies (e.g., `full`, `no_health`). |
| Render Mode | `None` | string or NoneType | $\{$`rgb_array`, `None`$\}$ | Rendering mode for visualization. |

## C.3 Reward Function Implementation Details

This appendix provides the detailed mathematical formulations for key reward functions implemented in the CONTAGIONRL environment. The goal of the agent, in the context of these reward functions, is to maximize the cumulative reward, which is designed to correlate with prolonging its susceptible state.

### C.3.1 Constant Reward

This is the simplest reward structure, providing a binary signal based on the agent's health state. Let $S_a$ denote the state of the agent. The reward $R_{\text{const}}$ at each timestep is defined as:

$$R_{\text{const}} = \begin{cases} 1 & \text{if } S_a = \text{Susceptible} \\ 0 & \text{otherwise} \end{cases} \tag{21}$$

### C.3.2 Reduce Infection Probability Reward

This reward function directly incentivizes the agent to minimize its own probability of becoming infected. Let $P_{\text{infect}}(a)$ be the agent's probability of infection at the current timestep, calculated considering its position, adherence, and proximity to infected individuals. The reward $R_{\text{reduce\_inf}}$ is defined as:

$$R_{\text{reduce\_inf}} = \begin{cases} (1 - P_{\text{infect}}(a))^2 & \text{if } S_a = \text{Susceptible} \\ -5 & \text{if } S_a \neq \text{Susceptible} \end{cases} \tag{22}$$

The quadratic term $(1 - P_{\text{infect}}(a))^2$ strongly rewards states with very low infection probability.

### C.3.3 Combined Reward

This function combines the incentive to reduce infection probability with a small constant bonus for remaining susceptible. Using $P_{\text{infect}}(a)$ as defined above, and $R_{\text{const}}$ (which is 1 if susceptible, 0 otherwise from the agent's perspective in this formulation if it's not infected):

$$R_{\text{combo}} = \begin{cases} 0.8 \cdot (1 - P_{\text{infect}}(a))^2 + 0.1 & \text{if } S_a = \text{Susceptible} \\ 0 & \text{if } S_a \neq \text{Susceptible} \end{cases} \tag{23}$$

The 0.1 term derives from $0.1 \times R_{\text{const}}$ where $R_{\text{const}} = 1$ when the agent is susceptible.

### C.3.4 Maximize Nearest Distance Reward

This reward encourages the agent to maintain a certain distance from other individuals, primarily those who are susceptible or infected. Let $d_{\min}$ be the Euclidean distance from the agent to the nearest human $h$ whose state $S_h \in \{\text{Susceptible, Infected}\}$. Let $D_\beta$ be the parameter maximum distance for beta calculation from the environment configuration. The reward $R_{\text{max\_dist}}$ is defined as:

$$R_{\text{max\_dist}} = \begin{cases} 0 & \text{if } S_a = \text{Infected} \\ 1 & \text{if } S_a \neq \text{Infected and no relevant humans (S or I) exist} \\ 1 & \text{if } S_a \neq \text{Infected and } d_{\min} \geq D_\beta \text{ (and } D_\beta > 0) \\ \max(0, d_{\min}/D_\beta) & \text{if } S_a \neq \text{Infected and } d_{\min} < D_\beta \text{ (and } D_\beta > 0) \end{cases} \tag{24}$$

If $D_\beta \leq 0$ (interpreted as infinite or disabled threshold in some contexts, though the code uses it as a divisor if positive), the behavior might implicitly rely on normalization or other logic not explicitly captured by this simplified $D_\beta > 0$ case. The provided code primarily gives reward based on $d_{\min}/D_\beta$ when $d_{\min} < D_\beta$ and $D_\beta > 0$.

### C.3.5 Potential Field Reward

The Potential Field reward function conceptualizes agent-human interactions as a system of forces, guiding the agent's movement based on the proximity and state of other individuals. It is a composite reward with several terms:

$$R_{\text{PF}} = w_{\text{health}} r_{\text{health}} + w_{\text{adherence}} r_{\text{adherence}} + w_{\text{movement}} r_{\text{move}} \tag{25}$$

In the implementation, the weights are $w_{\text{health}} = 0.1$, $w_{\text{adherence}} = 0.2$, and $w_{\text{movement}} = 0.7$.

**1. Health Reward ($r_{\text{health}}$):** This is a binary reward for maintaining a susceptible state.

$$r_{\text{health}} = \begin{cases} 1 & \text{if } S_a = \text{Susceptible} \\ 0 & \text{otherwise} \end{cases} \tag{26}$$

(This component can be ablated to 0 via the no health variant).

**2. Adherence Reward ($r_{\text{adherence}}$):** This reward is directly proportional to the agent's NPI adherence level. Let $\alpha_a$ be the agent's current adherence (a value in $[0, 1]$).

$$r_{\text{adherence}} = \alpha_a \tag{27}$$

(This component can be ablated to 0 via the no adherence variant).

**3. Movement Reward ($r_{\text{move}}$):** This component rewards the agent for moving in alignment with a suggested force vector $\mathbf{F}$ and optionally for matching its magnitude. Let the agent's current position be $\mathbf{p}_a = (x_a, y_a)$ and human $j$'s position be $\mathbf{p}_j = (x_j, y_j)$. The shortest displacement vector on the toroidal grid from human $j$ to the agent is $\Delta \mathbf{p}_j = (\Delta x_j, \Delta y_j)$, where:

$$\Delta x_j = (x_a - x_j + G/2) \pmod{G} - G/2 \tag{28}$$

$$\Delta y_j = (y_a - y_j + G/2) \pmod{G} - G/2 \tag{29}$$

Here, $G$ is the grid size. The squared distance is $d_j^2 = (\Delta x_j)^2 + (\Delta y_j)^2 + \epsilon_{\text{dist}}$, where $\epsilon_{\text{dist}}$ is a small constant to prevent division by zero.

The force contribution from each human $j$ depends on its state:

$$
\text{weight}_j = \begin{cases} W_I & \text{if human } j \text{ is Infected (I)} \\ W_S & \text{if human } j \text{ is Susceptible (S) (can be 0 in no Susceptible ablation)} \\ 0 & \text{otherwise} \end{cases} \tag{30}
$$

The scaling factor for the force from human $j$ is $\text{scale}_j = \text{weight}_j / (d_j^2)^{p/2}$. The implementation uses $p = 1$, $W_I = 1.0$, $W_S = 0.5$. The resultant force vector is $\mathbf{F} = (F_x, F_y) = \sum_j \text{scale}_j \Delta \mathbf{p}_j$. The normalized force direction is $\hat{\mathbf{F}} = \mathbf{F}/(\|\mathbf{F}\| + \epsilon_{\text{norm}})$.

Let the agent's chosen movement vector from the last action be $\mathbf{a} = (a_x, a_y)$. The directional alignment reward is:

$$
r_{\text{dir}} = \text{clip}\left( \frac{\mathbf{a} \cdot \hat{\mathbf{F}}}{\|\mathbf{a}\| + \epsilon_{\text{norm}}}, -1, 1 \right) \tag{31}
$$

The magnitude matching reward is:

$$
r_{\text{mag}} = \text{clip}(1 - |\|\mathbf{a}\| - \min(\|\mathbf{F}\|, 1.0)|, -1, 1) \tag{32}
$$

The combined movement reward is (by default, with $\beta_m = 0.25$):

$$
r_{\text{move}} = (1 - \beta_m)r_{\text{dir}} + \beta_m r_{\text{mag}} \tag{33}
$$

This term is subject to ablations:

- *No Magnitude*: $r_{\text{move}} = r_{\text{dir}}$

- *No Direction*: $r_{\text{move}} = r_{\text{mag}}$

- *No Movement*: $r_{\text{move}} = 0$

The small constants $\epsilon_{\text{dist}}$ and $\epsilon_{\text{norm}}$ $(10^{-8})$ are used to ensure numerical stability.

### C.4 Reward Function Information Scope

The summation in $\mathbf{F}$ iterates over all humans in the simulation irrespective of the agent's visibility radius parameter. Consequently, under POMDP configurations where the agent's observation is restricted to a radius $r$, the force vector $\mathbf{F}$ still incorporates contributions from humans beyond $r$. The observation function zeroes out features for humans outside the visibility radius, but the reward computation does not apply this filter. This asymmetry means the directional reward acts as an oracle-like gradient signal during training, informing the agent whether its action aligned with the globally optimal escape direction even when the agent could not observe all contributing humans. This is consistent with asymmetric actor-critic approaches where training-time information may exceed test-time observations (Pinto et al., 2017).

Table 6: Information scope of each reward function and its components. *Local* terms depend only on the agent's own state or action. *Global* terms are computed from non-agent humans regardless of the agent's visibility radius (under the default MDP configuration with `visibility_radius` $= -1$).

| Reward Function | Component | Scope | Depends On |
|---|---|---|---|
| Constant | Alive bonus | Local | Agent state ($S_a$) |
| Reduce Infection | Alive check | Local | Agent state ($S_a$) |
| | $(1 - P_{\text{inf}})^2$ | Global | Agent pos., adherence, all infected positions |
| Const. + Red. Inf. | Alive bonus | Local | Agent state ($S_a$) |
| | Alive check | Local | Agent state ($S_a$) |
| | $(1 - P_{\text{inf}})^2$ | Global | Agent pos., adherence, all infected positions |
| Max Near. Dist. | Alive check | Local | Agent state ($S_a$) |
| | $d_{\text{min}}/D_\beta$ | Global | Agent pos., all S & I positions |
| Potential Field | Health ($r_{\text{health}}$) | Local | Agent state ($S_a$) |
| | Adherence ($r_{\text{adherence}}$) | Local | Agent action ($\alpha$) |
| | Direction ($r_{\text{dir}}$) | Global | Agent action, all S & I positions |
| | Magnitude ($r_{\text{mag}}$) | Global | Agent action, all S & I positions |

### C.5 Algorithmic Comparison Hyperparameters

The following tables summarize the hyperparameters used for the Proximal Policy Optimization (PPO), Soft Actor-Critic (SAC), and Advantage Actor-Critic (A2C) algorithms in our comparative study (Figure 2).

Table 7: PPO Hyperparameters

| Hyperparameter | Value |
|---|---|
| Policy Type | `MultiInputPolicy` |
| *Policy Kwargs:* | |
|    Net Arch (pi) | `[256, 256, 256, 256]` |
|    Net Arch (vf) | `[256, 256, 256, 256]` |
|    Activation Fn | `ReLU` |
|    Ortho Init | `True` |
| Batch Size | 2048 |
| N Steps | 1024 |
| N Epochs | 5 |
| Learning Rate | `3e-4` |
| Gamma | 0.96 |
| GAE Lambda | 0.95 |
| Target KL | 0.04 |
| Clip Range | 0.2 |
| Ent Coef | 0.02 |
| Normalize Advantage | `True` |
| Total Timesteps | 8,000,000 |
| N Envs | 4 |

Table 8: SAC Hyperparameters

| Hyperparameter | Value |
|---|---|
| Policy Type | MlpPolicy |
| *Policy Kwargs:* | |
|    Net Arch (pi) | [256, 256, 256, 256] |
|    Net Arch (qf) | [256, 256, 256, 256] |
|    Activation Fn | ReLU |
| Learning Rate | 3e-4 |
| Buffer Size | 1,000,000 |
| Batch Size | 256 |
| Tau | 0.005 |
| Train Freq | 1 |
| Gradient Steps | 1 |
| Ent Coef | auto |
| Gamma | 0.96 |
| Total Timesteps | 8,000,000 |
| N Envs | 4 |

Table 9: A2C Hyperparameters

| Hyperparameter | Value |
|---|---|
| Policy Type | MlpPolicy |
| *Policy Kwargs:* | |
|    Net Arch (pi) | [256, 256, 256, 256] |
|    Net Arch (vf) | [256, 256, 256, 256] |
|    Activation Fn | ReLU |
|    Ortho Init | True |
| N Steps | 640 |
| Gamma | 0.96 |
| GAE Lambda | 0.95 |
| Ent Coef | 0.01 |
| VF Coef | 0.5 |
| Max Grad Norm | 0.5 |
| Learning Rate | 3e-4 |
| Use RMS Prop | True |
| Normalize Advantage | True |
| Total Timesteps | 8,000,000 |
| N Envs | 4 |

## C.6   Baseline Implementation Details

This section provides further detail on the implementation of the non-learning baseline policies used for comparative analysis, as described in Section 4.

### C.6.1   Stationary Policy Implementation

The Stationary policy represents a minimal activity baseline. Its implementation is deterministic: at every timestep $t$ within an episode, the agent selects a constant action vector $a(t) = [0.0, 0.0, 0.0]$. This corresponds to zero intended movement ($\Delta x = 0, \Delta y = 0$) and zero NPI adherence ($\alpha = 0$). The agent's position remains fixed, and its susceptibility is solely determined by the base infection rate $\beta$ and its exposure to infected individuals based on the environment dynamics.

### C.6.2 Random Policy Implementation

The Random policy simulates behavior devoid of environmental feedback or strategy. At each timestep $t$, the action $a(t) = (\Delta x, \Delta y, \alpha)$ is generated by sampling independently from uniform distributions:

- Movement components $\Delta x, \Delta y \sim \mathcal{U}(-1, 1)$.

- The adherence level $\alpha \sim \mathcal{U}(0, 1)$.

This ensures that the agent explores the action space randomly without any directed objective, using the simulation's internal pseudo-random number generator for consistency.

### C.6.3 Greedy Distance Maximizer Policy Implementation

The Greedy policy enacts a deterministic, reactive heuristic focused on maximizing immediate distance from the nearest identified threat. The implementation logic proceeds as follows at each timestep $t$:

1. **Identify Nearest Threat:** Let the agent's position at time $t$ be $\mathbf{p}_a(t) = (x_a(t), y_a(t))$. Identify the set of currently infected humans $\mathcal{I}(t)$. This step utilizes privileged knowledge of the true state $S_j \in \{S, I, R, D\}$ for all humans $j$. If $\mathcal{I}(t)$ is empty, the agent defaults to a stationary action $\mathbf{m}^* = (0, 0)$.

2. **Target Selection:** If $\mathcal{I}(t)$ is not empty, calculate the Euclidean distance $d(\mathbf{p}_a(t), \mathbf{p}_j(t))$ for all $j \in \mathcal{I}(t)$, where $\mathbf{p}_j(t)$ is the position of infected human $j$ and $d(\cdot, \cdot)$ accounts for the toroidal grid geometry. Identify the single infected human $h_{\text{nearest}}(t)$ corresponding to the minimum distance:

$$h_{\text{nearest}}(t) = \arg\min_{j \in \mathcal{I}(t)} d(\mathbf{p}_a(t), \mathbf{p}_j(t))$$

   The exact position $\mathbf{p}_{h_{\text{nearest}}}(t)$ is **privileged information**.

3. **Evaluate Potential Moves:** Define a discrete set of candidate movement vectors $\mathcal{A}_{\text{move}}$. This set includes the zero vector $(0, 0)$ and scaled unit vectors representing the maximum possible step in the eight cardinal and diagonal directions, e.g., $\{(0, 0), (\pm s_M, 0), (0, \pm s_M), (\pm s_M/\sqrt{2}, \pm s_M/\sqrt{2})\}$, where $s_M$ is the maximum movement scale (typically 1.0).

4. **Select Best Move:** For each candidate movement $\mathbf{m}_i = (\Delta x_i, \Delta y_i) \in \mathcal{A}_{\text{move}}$, calculate the agent's potential next position $\mathbf{p}'_{a,i}(t+1)$ by applying the movement to $\mathbf{p}_a(t)$ and considering the grid's periodic boundaries. Evaluate the distance from this potential position to the initially identified nearest threat's current position $\mathbf{p}_{h_{\text{nearest}}}(t)$. Select the movement vector $\mathbf{m}^*$ that maximizes this distance:

$$\mathbf{m}^* = \arg\max_{\mathbf{m}_i \in \mathcal{A}_{\text{move}}} d(\mathbf{p}'_{a,i}(t+1), \mathbf{p}_{h_{\text{nearest}}}(t))$$

   This calculation relies on knowing $\mathbf{p}_{h_{\text{nearest}}}(t)$.

5. **Set Adherence:** The adherence component of the action is deterministically set to the maximum value, $\alpha = 1.0$.

6. **Final Action:** The resulting action for timestep $t$ is $a(t) = (\mathbf{m}^*, \alpha = 1.0)$.

This implementation defines a simple, reactive strategy that exploits complete and accurate environmental state information to maximize instantaneous separation from the nearest perceived threat, while also employing maximum protective adherence.

# D    Results Continued

These figures and tables provide additional visualizations and experiments corresponding to those presented in the main Results section of the paper.

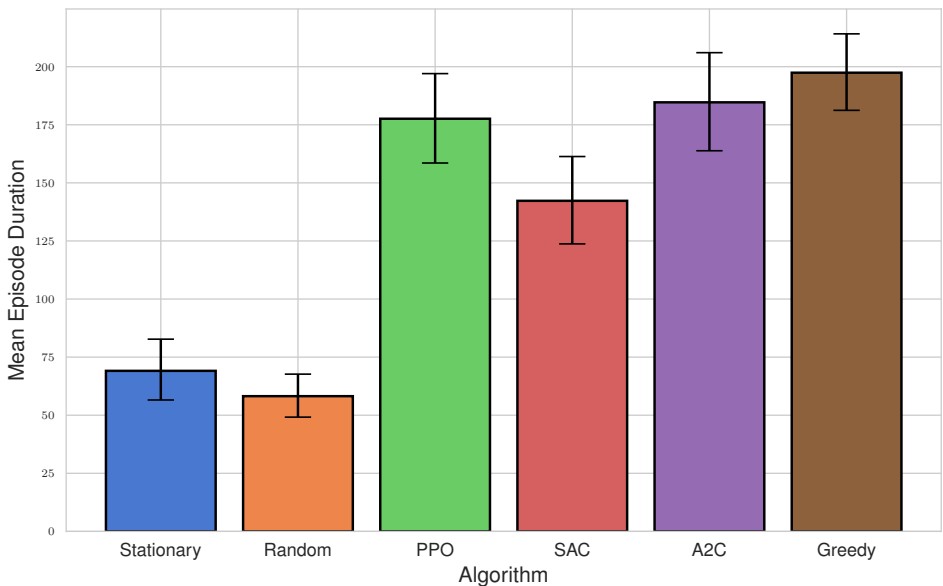

Figure 9: Mean episode durations and 95% bootstrapped confidence intervals (10,000 samples) for each agent. Values are aggregated across 3 training seeds and 100 evaluation episodes per seed. This figure summarizes the performance trends visualized in Figure 2.

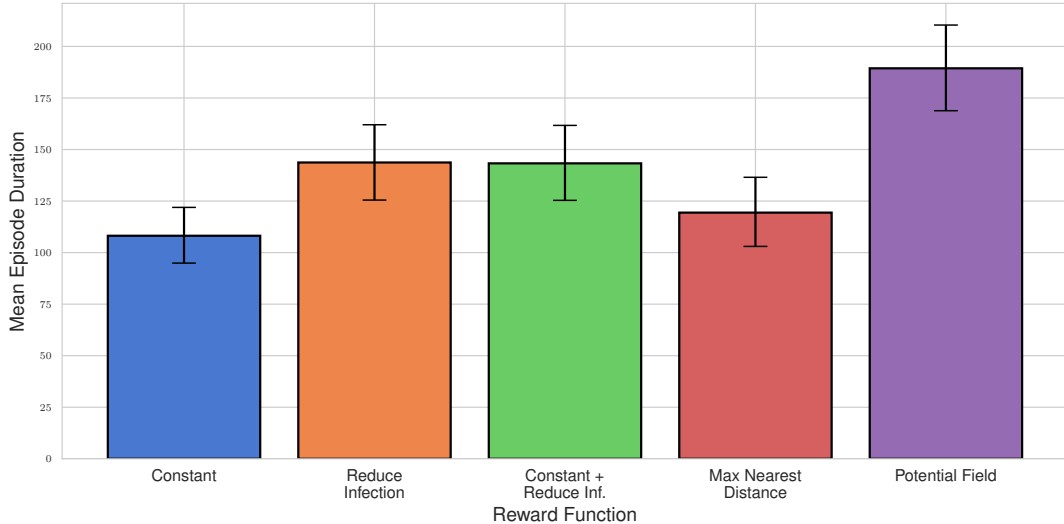

Figure 10: Mean episode durations with 95% bootstrapped confidence intervals (10,000 samples) for each reward function. These values correspond to the same experiments shown in Figure 3, aggregating performance across 3 seeds and 100 evaluations per seed.

Table 10: One-sided Mann–Whitney U test results comparing each reward function to the *Potential Field* baseline (PF). Bonferroni correction is applied to one-sided $p$-values. "Winner" denotes the reward function with significantly longer episode durations. Abbreviations: **Const** = Constant, **RedInfect** = Reduce Infection, **Combo** = Constant + Reduce Infection, **MaxDist** = Max Nearest Distance, **PF** = Potential Field. Significance: * $p < 0.05$, ** $p < 0.01$, *** $p < 0.001$, **n.s.** = not significant. This table supports the significance annotations shown in Figure 3.

| Reward Function | $p$ (2-sided) | $p$ (1-sided) | Sig (2) | Corrected $p$ (1) | Sig (1) | Winner |
|---|---|---|---|---|---|---|
| Const | 3.38e-06 | 1.69e-06 | *** | 6.76e-06 | *** | PF |
| RedInfect | 0.00556 | 0.00278 | ** | 0.0111 | * | PF |
| Combo | 0.00185 | 0.00092 | ** | 0.00369 | ** | PF |
| MaxDist | 1.04e-06 | 5.21e-07 | *** | 2.08e-06 | *** | PF |

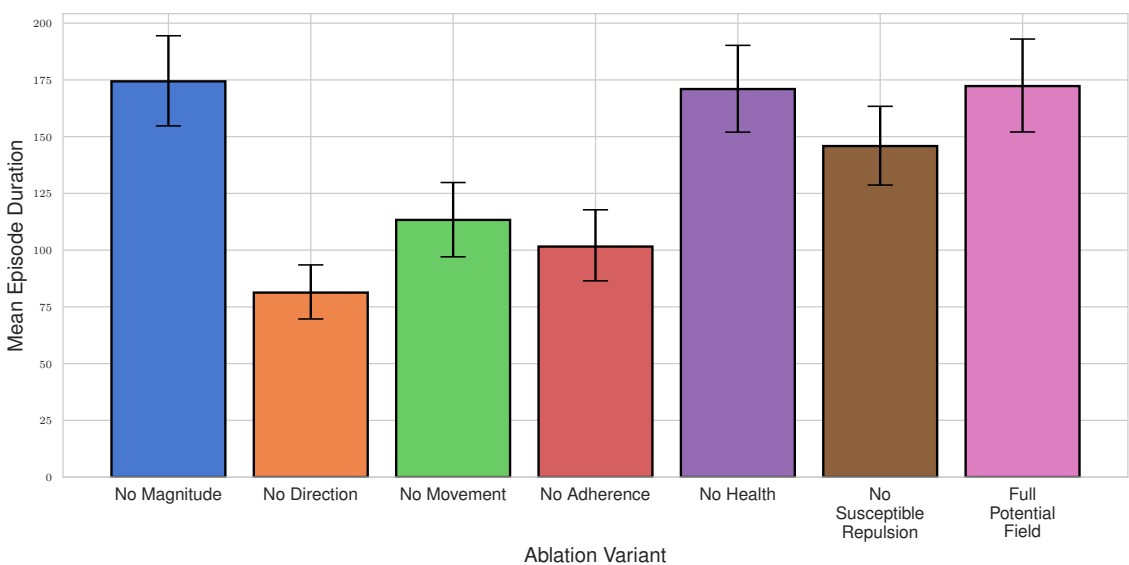

Figure 11: Mean episode durations and 95% bootstrapped confidence intervals (10,000 samples) for each ablation variant in the Potential Field reward study. These aggregate values summarize the same evaluations shown in Figure 4, providing a direct comparison of central performance tendencies across reward configurations.

Table 11: One-sided Mann–Whitney U test results comparing each ablation variant to the *Full Potential Field* (**FPF**), which represents the original, unmodified reward function. Bonferroni correction is applied to one-sided $p$-values. "Winner" indicates the variant with significantly longer episode durations. Significance levels: * $p < 0.05$, ** $p < 0.01$, *** $p < 0.001$, **n.s.** = not significant. Results support the differences shown in Figure 4.

| Ablation Variant | p (2-sided) | p (1-sided) | Sig (2) | p (1) Corr | Sig (1) | Winner |
|---|---|---|---|---|---|---|
| No Magnitude | 0.9571 | 0.5216 | n.s. | 1 | n.s. | − |
| No Direction | 3.80e-11 | 1.90e-11 | *** | 1.14e-10 | *** | FPF |
| No Movement | 6.93e-06 | 3.46e-06 | *** | 2.08e-05 | *** | FPF |
| No Adherence | 3.88e-09 | 1.94e-09 | *** | 1.16e-08 | *** | FPF |
| No Health | 0.8242 | 0.5881 | n.s. | 1 | n.s. | − |
| No Susceptible Repulsion | 0.1336 | 0.0668 | n.s. | 0.4007 | n.s. | − |

Table 12: Statistical significance tests comparing RL agents trained with different visibility radius constraints in epidemic control. Agents were trained and evaluated with Full Visibility, Limited Visibility r=10, r=15, and r=20 (where r represents the radius within which the agent can observe infected individuals). Each condition used 300 episodes (3 seeds × 100 episodes per seed). Statistical significance determined using Mann-Whitney U tests with Bonferroni correction for 18 multiple comparisons (corrected $\alpha = 0.002778$). Tests compare trained agents against baselines (Stationary, Random, Greedy) and trained agents against each other to assess the impact of visibility constraints on performance.

| Category | Comparison | Winner | U-stat. | p-value | Sig. |
|---|---|---|---|---|---|
| Trained vs Baselines | Full vs Stationary | Full | 79,732.0 | $p < 0.001$ | *** |
| | Full vs Random | Full | 66,639.0 | $p < 0.001$ | *** |
| | Full vs Greedy | Full | 46,927.0 | $p = 1.000$ | n.s. |
| | r=10 vs Stationary | r=10 | 82,063.0 | $p < 0.001$ | *** |
| | r=10 vs Random | r=10 | 73,277.0 | $p < 0.001$ | *** |
| | r=10 vs Greedy | r=10 | 55,948.0 | $p < 0.001$ | *** |
| | r=15 vs Stationary | r=15 | 83,000.0 | $p < 0.001$ | *** |
| | r=15 vs Random | r=15 | 74,393.0 | $p < 0.001$ | *** |
| | r=15 vs Greedy | r=15 | 59,913.0 | $p < 0.001$ | *** |
| | r=20 vs Stationary | r=20 | 82,436.0 | $p < 0.001$ | *** |
| | r=20 vs Random | r=20 | 72,773.0 | $p < 0.001$ | *** |
| | r=20 vs Greedy | r=20 | 57,229.0 | $p < 0.001$ | *** |
| Trained vs Trained | Full vs r=10 | r=10 | 35,009.0 | $p < 0.001$ | *** |
| | Full vs r=15 | r=15 | 31,504.0 | $p < 0.001$ | *** |
| | Full vs r=20 | r=20 | 34,320.0 | $p < 0.001$ | *** |
| | r=10 vs r=15 | r=15 | 40,943.0 | $p = 1.000$ | n.s. |
| | r=10 vs r=20 | r=20 | 43,770.0 | $p = 1.000$ | n.s. |
| | r=15 vs r=20 | r=15 | 47,696.0 | $p = 1.000$ | n.s. |

# E   Learned Adherence Behavior Analysis

To investigate adherence usage of trained models across different reward formulations, we conducted an analysis of the learned adherence behaviors. This analysis examines if agents can independently discover the value of non-pharmaceutical intervention (NPI) compliance or if they require explicit reward tuning for discovering the importance of adherence.

**Methodology.** We measured the mean and standard deviation of adherence actions from trained policies across all reward functions and potential field ablation variants. Each policy was evaluated using the same experimental setup as the main results. Adherence values were recorded at each timestep and aggregated across episodes.

**Results.** Table 13 shows the learned adherence usage across reward functions. Agents trained with simple reward functions like Constant, Reduce Infection, and Constant + Reduce Infection achieve maximal adherence ($\alpha = 1.0$) with minimal variance. This indicates that even without explicit adherence incentives, temporal credit assignment enables agents to discover the survival benefit of full NPI compliance through the sparse reward signal. The Max Distance reward function yields slightly lower mean adherence with higher variance, suggesting that spatial optimization partially competes with adherence learning in this formulation.

The Full Potential Field reward achieves maximal adherence, consistent with its explicit adherence component. However, when the adherence reward is removed from potential field reward function (PF: No Adherence), mean adherence drops to 0.41 with high variance of 0.47. This indicates essentially random adherence behavior. All other potential field ablations (No Direction, No Health, No Magnitude, No Movement and No Susceptible Repulsion) maintain maximal adherence despite their respective component removals.

These results reveal an **asymmetry in adherence learnability based on reward complexity**. Simple reward functions allow agents to implicitly learn maximal adherence through the survival signal alone. However, in complex multi-component objectives like the potential field reward, agents fail to discover adherence value without explicit incentivization. The diverse reward signals in the potential field formulation appear to dilute the implicit survival gradient that simpler rewards preserve.

The ContagionRL platform includes a configurable adherence penalty parameter (Table 5) to enable the future investigation of strategic trade-offs where agents must balance survival benefits against adherence costs.

Table 13: Learned adherence behavior across reward functions and potential field ablations. Mean and standard deviation computed over 300 evaluation episodes (3 seeds $\times$ 100 episodes). PF denotes Potential Field.

| Reward Function | Mean Adherence | Std Adherence |
|---|---|---|
| Constant | 1.000 | 0.000 |
| Reduce Infection | 1.000 | 0.000 |
| Constant + Reduce Inf | 1.000 | 0.000 |
| Max Distance | 0.872 | 0.312 |
| Potential Field | 1.000 | 0.001 |
| PF: No Adherence | 0.406 | 0.472 |
| PF: No Direction | 1.000 | 0.000 |
| PF: No Health | 1.000 | 0.000 |
| PF: No Magnitude | 1.000 | 0.000 |
| PF: No Move | 1.000 | 0.000 |
| PF: No S Repulsion | 1.000 | 0.001 |

## F Partial Observability Feature Analysis

To further understand the counter-intuitive result that limited visibility agents outperform full visibility agents (Figure 5), we conducted a feature importance analysis measuring how strongly each trained model responds to observed human positions.

**Methodology.** We computed Spearman correlations ($\rho$) between the mean relative positions of visible humans and the agent's movement actions. Specifically, we measured correlations between the mean x-displacement of observed humans and the agent's x-movement ($\rho_{\Delta x}$), and similarly for the y-dimension ($\rho_{\Delta y}$).

**Results.** Table 14 presents the action-position correlations by visibility radius. All correlations are negative, indicating avoidance behavior where agents move away from observed humans. Limited visibility models exhibit substantially stronger correlations than full visibility models. Agents trained with r=10 visibility show correlations approximately 2-2.5$\times$ stronger than full visibility agents ($p < 0.001$), demonstrating more decisive responses to observed threats.

**Interpretation.** The stronger correlations under limited visibility occur because full visibility dilutes attention across all humans, many of whom are distant and pose negligible infection risk. Limited visibility effectively filters the observation space to include only nearby individuals who have a larger impact on the effective transmission rate since transmission follows distance decay.

Limited visibility thus acts as an implicit attention mechanism, restricting observations to the radius where humans actually pose significant threat. Full visibility forces agents to process information about distant humans who have near-zero impact on infection risk, increasing observation dimensionality without proportional informational benefit. These results suggest that appropriately scoped observations matched to the task's distance-dependent mechanics produce more focused and effective policies, rather than requiring more expressive architectures to handle larger state spaces.

Table 14: Spearman correlations between mean relative positions of visible humans and agent movement actions across visibility conditions. Negative values indicate avoidance behavior. All correlations significant at $p < 0.001$.

| Visibility | $\rho_{\Delta x}$ | $\rho_{\Delta y}$ |
|---|---|---|
| Limited (r=10) | -0.557 | -0.540 |
| Limited (r=15) | -0.441 | -0.464 |
| Limited (r=20) | -0.342 | -0.438 |
| Full Visibility | -0.220 | -0.234 |

## G Compute Resources

All experiments were conducted on Apple Silicon M4 chip workers (16-core CPU) with 48 GB LPDDR5 unified memory. We devoted approximately 40 CPU-hours to hyperparameter optimization. All training and evaluation were performed across 3 random seeds (with 100 runs per seed unless otherwise specified). Training times for each set of experiments were approximately as follows:

- Comparison of different RL algorithms: 22 hours

- Ablation studies of the potential field reward function: 13 hours

- Comparison of reward functions: 14 hours

- Comparison of POMDP and full observability (MDP): 15 hours

- Constraints in Movement: 12 hours

- Additional results and experiments (in appendix): 26 hours

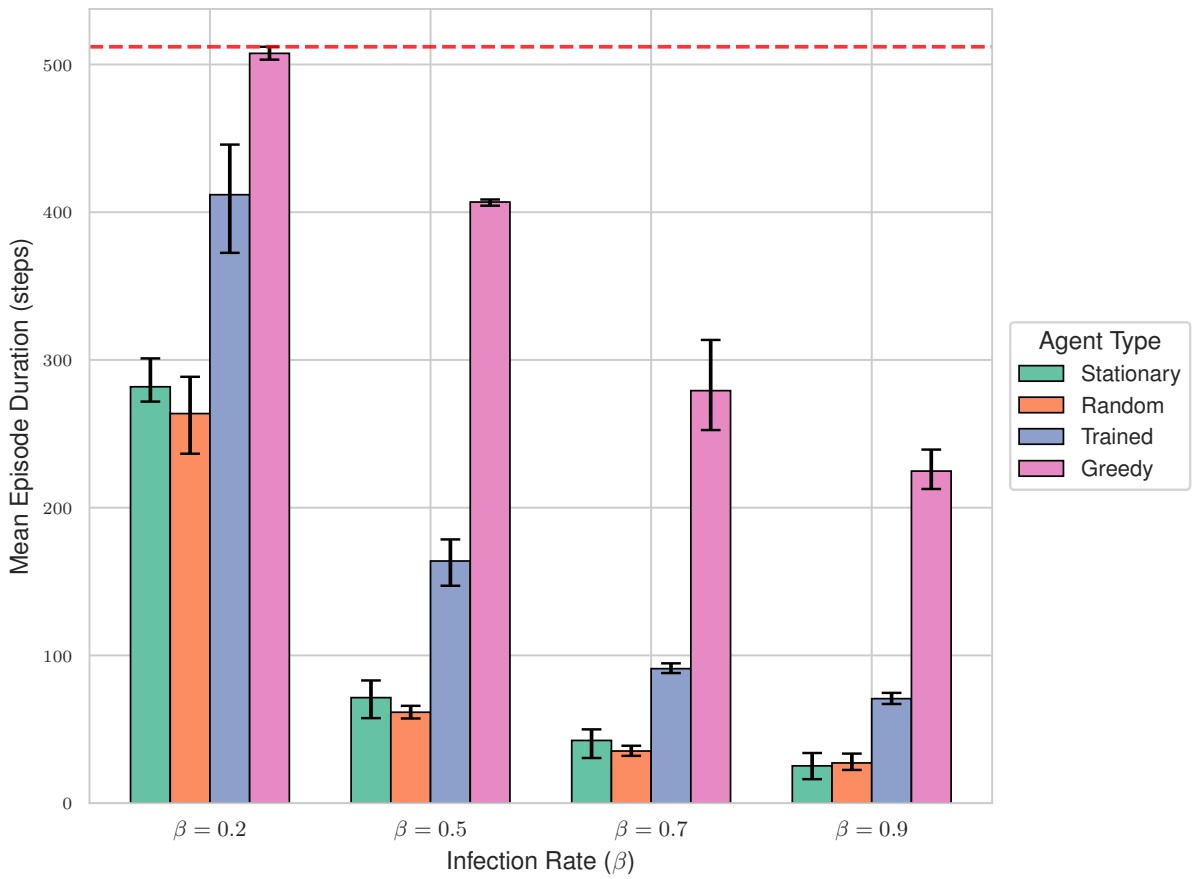

Figure 12: Comparison of mean episode durations across agents (Trained, Stationary, Random, Greedy) for different infection rates $\beta$. Bars indicate the mean episode duration over 3 seeds (each evaluated for 100 episodes, totaling 300 episodes per agent), with 95% bootstrapped confidence intervals (10,000 resamples). The dashed red line marks the maximum possible episode duration. See Table 15 for detailed two-sided and one-sided Mann–Whitney U test results with Bonferroni correction.

## H  Additional Results and Experiments

In the supplementary experiments presented here, we vary one environmental parameter of the epidemic at a time to simulate a range of spatial epidemic scenarios constructible with CONTAGIONRL. This allows us to explore both favorable and challenging conditions under which to stress-test agent learning. For each experiment, we provide two outputs: a grouped bar plot illustrating mean episode durations with 95% confidence intervals, and a table reporting results from one-sided and two-sided Mann–Whitney U tests. All experiments in this section use PPO for training, with hyperparameters defined in Appendix C.5 and environment configurations in Appendix C.2. Each table and figure corresponds to a single modified environmental parameter specified in their respective captions.

Table 15: Results of two-sided and one-sided Mann–Whitney U tests comparing **Trained** agents to baselines (**Stationary**, **Random**, **Greedy**) across four infection rates ($\beta$). Each comparison uses 300 evaluation episodes (100 per seed over 3 seeds). The one-sided $p$-values are Bonferroni-corrected within each $\beta$ group. "Winner" indicates the agent with significantly longer episode durations based on the corrected one-sided test. Significance levels: * $p < 0.05$, ** $p < 0.01$, *** $p < 0.001$, **n.s.** = not significant.

| $\beta$ | Baseline | $p$ (2-sided) | $p$ (1-sided) | Sig (2) | $p$ (1) Corr | Sig (1) | Winner |
|---|---|---|---|---|---|---|---|
| | Stationary | $2.86 \times 10^{-17}$ | $1.43 \times 10^{-17}$ | *** | $4.30 \times 10^{-17}$ | *** | Trained |
| 0.2 | Random | $6.87 \times 10^{-24}$ | $3.43 \times 10^{-24}$ | *** | $1.03 \times 10^{-23}$ | *** | Trained |
| | Greedy | $2.03 \times 10^{-21}$ | $1.02 \times 10^{-21}$ | *** | $3.05 \times 10^{-21}$ | *** | Greedy |
| | Stationary | $1.01 \times 10^{-16}$ | $5.06 \times 10^{-17}$ | *** | $1.52 \times 10^{-16}$ | *** | Trained |
| 0.5 | Random | $5.13 \times 10^{-15}$ | $2.57 \times 10^{-15}$ | *** | $7.70 \times 10^{-15}$ | *** | Trained |
| | Greedy | $1.29 \times 10^{-46}$ | $6.45 \times 10^{-47}$ | *** | $1.94 \times 10^{-46}$ | *** | Greedy |
| | Stationary | $9.82 \times 10^{-16}$ | $4.91 \times 10^{-16}$ | *** | $1.47 \times 10^{-15}$ | *** | Trained |
| 0.7 | Random | $2.37 \times 10^{-12}$ | $1.18 \times 10^{-12}$ | *** | $3.55 \times 10^{-12}$ | *** | Trained |
| | Greedy | $1.56 \times 10^{-27}$ | $7.79 \times 10^{-28}$ | *** | $2.34 \times 10^{-27}$ | *** | Greedy |
| | Stationary | $5.46 \times 10^{-27}$ | $2.73 \times 10^{-27}$ | *** | $8.19 \times 10^{-27}$ | *** | Trained |
| 0.9 | Random | $2.53 \times 10^{-15}$ | $1.27 \times 10^{-15}$ | *** | $3.80 \times 10^{-15}$ | *** | Trained |
| | Greedy | $6.74 \times 10^{-23}$ | $3.37 \times 10^{-23}$ | *** | $1.01 \times 10^{-22}$ | *** | Greedy |

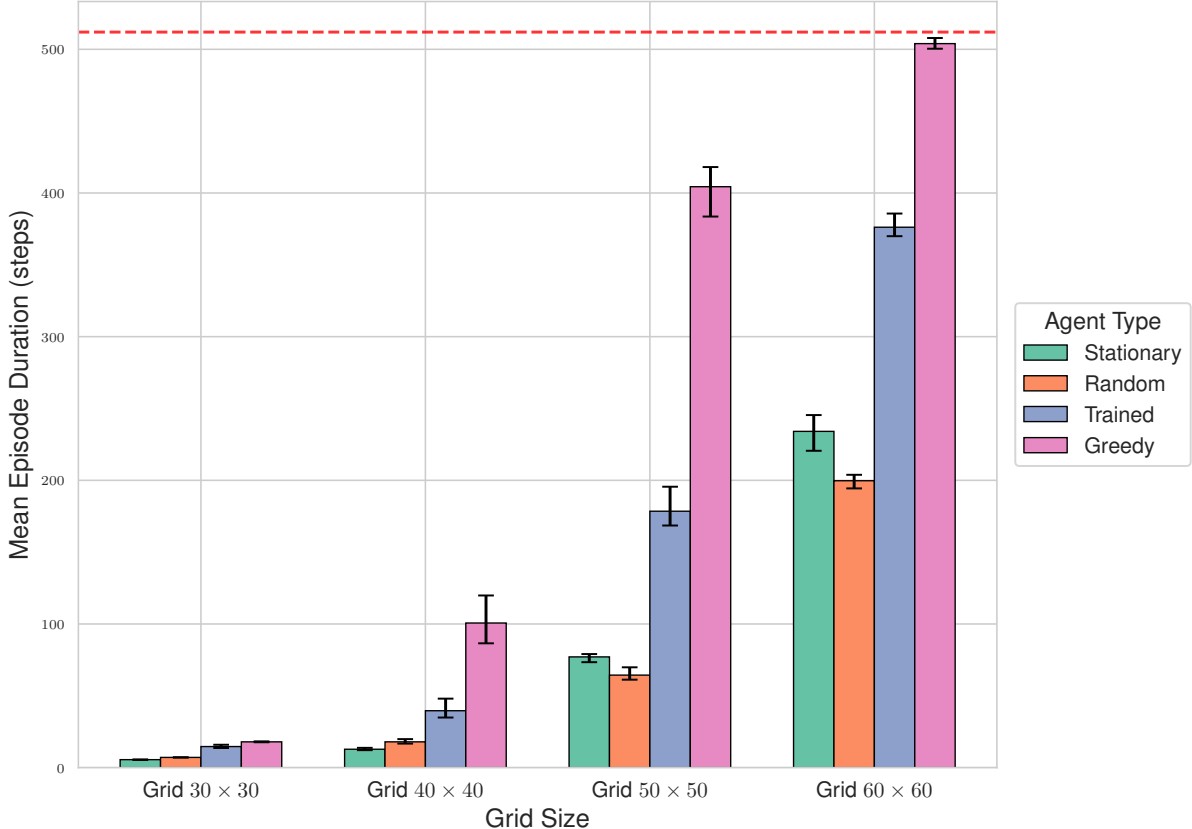

Figure 13: Mean episode durations for *Trained*, *Stationary*, *Random*, and *Greedy* agents evaluated across environments with varying grid sizes. Each bar represents the mean across 3 seeds (each with 100 evaluation episodes), and is accompanied by 95% bootstrapped confidence intervals (10,000 resamples). The red dashed line marks the maximum episode length. See Table 16 for statistical comparisons between *Trained* and baseline agents using both one-sided and two-sided Mann–Whitney U tests with Bonferroni correction.

Table 16: One-sided and two-sided Mann–Whitney U test results comparing *Trained* agents to baseline agents across varying grid sizes. For each grid configuration, we tested whether the *Trained* agent achieves significantly longer episode durations than *Stationary*, *Random*, and *Greedy* baselines. Bonferroni correction is applied to the one-sided $p$-values. Significance levels: * $p < 0.05$, ** $p < 0.01$, *** $p < 0.001$, **n.s.** = not significant. "Winner" indicates the model with significantly longer duration after correction. "Mean Diff" refers to (Trained – Baseline) mean.

| Grid | Baseline | $p$ (2-sid.) | $p$ (1-sid.) | Sig (2) | $p$ (1) Corr | Sig (1) | Winner | Mean Diff |
|---|---|---|---|---|---|---|---|---|
| $30 \times 30$ | Stationary | 1.96e-40 | 9.80e-41 | *** | 2.94e-40 | *** | Trained | 9.12 |
| | Random | 1.62e-24 | 8.13e-25 | *** | 2.44e-24 | *** | Trained | 7.56 |
| | Greedy | 0.1482 | 0.0741 | n.s. | 0.2223 | n.s. | − | −3.27 |
| $40 \times 40$ | Stationary | 3.11e-29 | 1.55e-29 | *** | 4.67e-29 | *** | Trained | 26.81 |
| | Random | 1.74e-17 | 8.71e-18 | *** | 2.61e-17 | *** | Trained | 21.66 |
| | Greedy | 6.47e-12 | 3.23e-12 | *** | 9.70e-12 | *** | Greedy | −61.05 |
| $50 \times 50$ | Stationary | 1.01e-16 | 5.06e-17 | *** | 1.52e-16 | *** | Trained | 101.40 |
| | Random | 4.04e-16 | 2.02e-16 | *** | 6.06e-16 | *** | Trained | 114.08 |
| | Greedy | 2.07e-37 | 1.04e-37 | *** | 3.11e-37 | *** | Greedy | −225.88 |
| $60 \times 60$ | Stationary | 7.98e-16 | 3.99e-16 | *** | 1.20e-15 | *** | Trained | 142.10 |
| | Random | 4.25e-24 | 2.13e-24 | *** | 6.38e-24 | *** | Trained | 176.46 |
| | Greedy | 1.12e-24 | 5.57e-25 | *** | 1.67e-24 | *** | Greedy | −127.80 |

Table 17: Mann–Whitney U test results comparing *Trained* agents to *Stationary*, *Random*, and *Greedy* baselines under varying adherence effectiveness. Both two-sided and one-sided tests were performed. One-sided $p$-values were Bonferroni-corrected. "Winner" indicates the agent with significantly longer episode duration. "Mean Diff" is Trained minus Baseline. Significance: * $p < 0.05$, ** $p < 0.01$, *** $p < 0.001$, **n.s.** = not significant.

| Adh. Eff. | Baseline | $p$ (2-sid.) | $p$ (1-sid.) | Sig (2) | $p$ (1) Corr | Sig (1) | Winner | Mean Diff |
|---|---|---|---|---|---|---|---|---|
| 0.1 | Stationary | 1.05e-35 | 5.25e-36 | *** | 1.58e-35 | *** | Trained | 172.85 |
| | Random | 4.14e-33 | 2.07e-33 | *** | 6.21e-33 | *** | Trained | 177.92 |
| | Greedy | 4.49e-34 | 2.24e-34 | *** | 6.73e-34 | *** | Greedy | −192.65 |
| 0.3 | Stationary | 1.07e-08 | 5.37e-09 | *** | 1.61e-08 | *** | Trained | 51.67 |
| | Random | 2.36e-06 | 1.18e-06 | *** | 3.55e-06 | *** | Trained | 62.49 |
| | Greedy | 1.22e-43 | 6.12e-44 | *** | 1.84e-43 | *** | Greedy | −249.12 |
| 0.5 | Stationary | 0.0180 | 0.0090 | * | 0.0270 | * | Trained | 25.03 |
| | Random | 6.83e-06 | 3.42e-06 | *** | 1.03e-05 | *** | Trained | 51.82 |
| | Greedy | 4.72e-42 | 2.36e-42 | *** | 7.07e-42 | *** | Greedy | −242.83 |

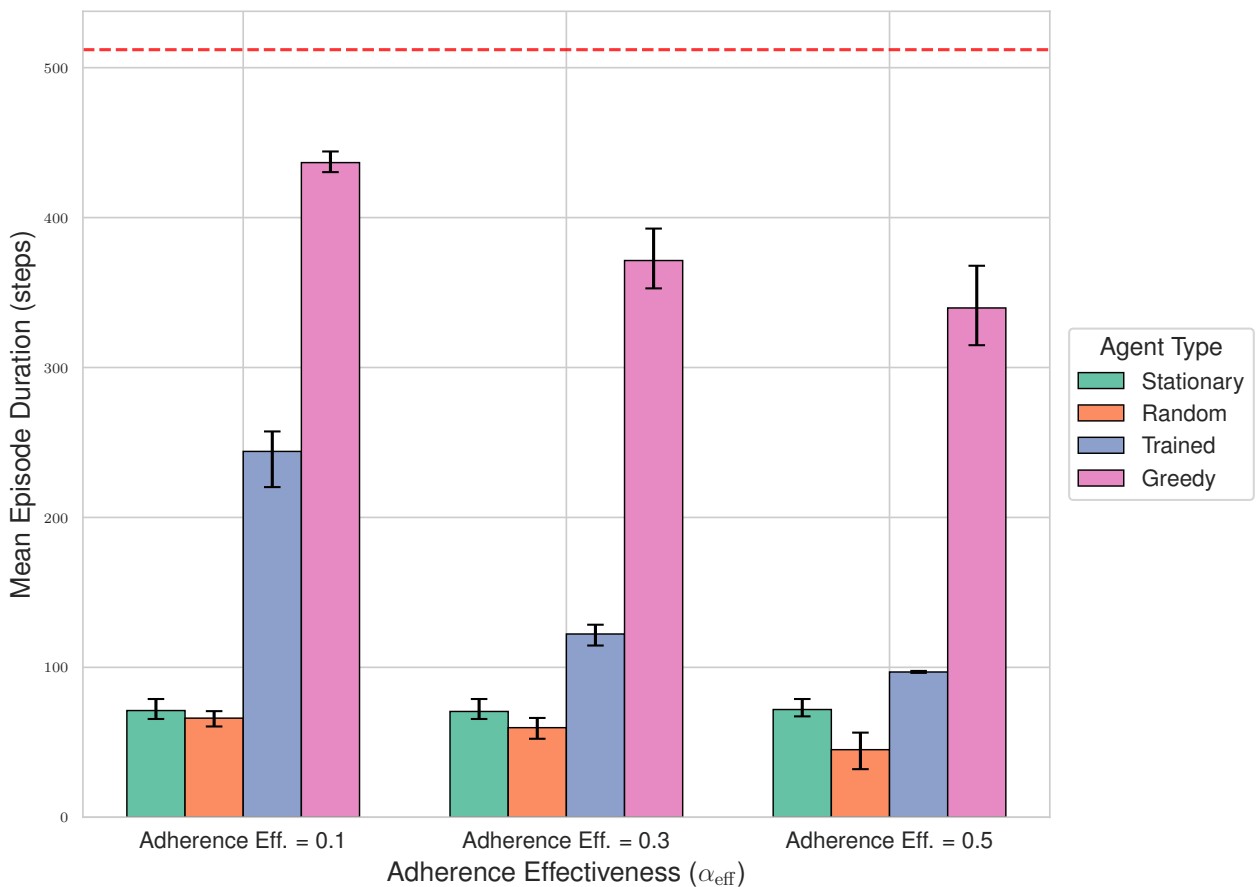

Figure 14: Mean episode durations of *Trained*, *Stationary*, *Random*, and *Greedy* agents across varying levels of adherence effectiveness. Each bar represents the mean of 3 independent training runs, each evaluated over 100 episodes (300 evaluations per agent-type per setting), with 95% bootstrapped confidence intervals (10,000 resamples). A red dashed line indicates the maximum episode length imposed by the environment. Statistical comparisons between the *Trained* agent and each baseline are provided in Table 17, using one-sided and two-sided Mann–Whitney U tests with Bonferroni correction.

Table 18: Mann–Whitney U test results comparing *Trained* agents to *Stationary*, *Random*, and *Greedy* baselines across varying levels of the *distance decay* parameter. Both two-sided and one-sided tests were performed. One-sided $p$-values were corrected using Bonferroni correction. "Winner" denotes the agent with significantly longer episode durations after correction. "Mean Diff" reports the difference in mean episode duration between *Trained* and baseline (positive = *Trained* better). Significance thresholds: * $p < 0.05$, ** $p < 0.01$, *** $p < 0.001$, **n.s.** = not significant.

| Dist. Decay | Baseline | $p$ (2-sid.) | $p$ (1-sid.) | Sig (2) | $p$ (1) Corr | Sig (1) | Winner | Mean Diff |
|---|---|---|---|---|---|---|---|---|
| 0.15 | Stationary | 2.42e-26 | 1.21e-26 | *** | 3.63e-26 | *** | Trained | 8.44 |
| | Random | 3.97e-18 | 1.99e-18 | *** | 5.96e-18 | *** | Trained | 17.90 |
| | Greedy | 6.77e-14 | 3.39e-14 | *** | 1.02e-13 | *** | Greedy | −96.67 |
| 0.30 | Stationary | 1.09e-24 | 5.44e-25 | *** | 1.63e-24 | *** | Trained | 113.78 |
| | Random | 1.73e-21 | 8.64e-22 | *** | 2.59e-21 | *** | Trained | 117.54 |
| | Greedy | 9.73e-35 | 4.86e-35 | *** | 1.46e-34 | *** | Greedy | −213.42 |
| 0.45 | Stationary | 2.91e-21 | 1.46e-21 | *** | 4.37e-21 | *** | Trained | 147.33 |
| | Random | 2.84e-28 | 1.42e-28 | *** | 4.26e-28 | *** | Trained | 165.01 |
| | Greedy | 1.42e-15 | 7.11e-16 | *** | 2.13e-15 | *** | Greedy | −77.05 |

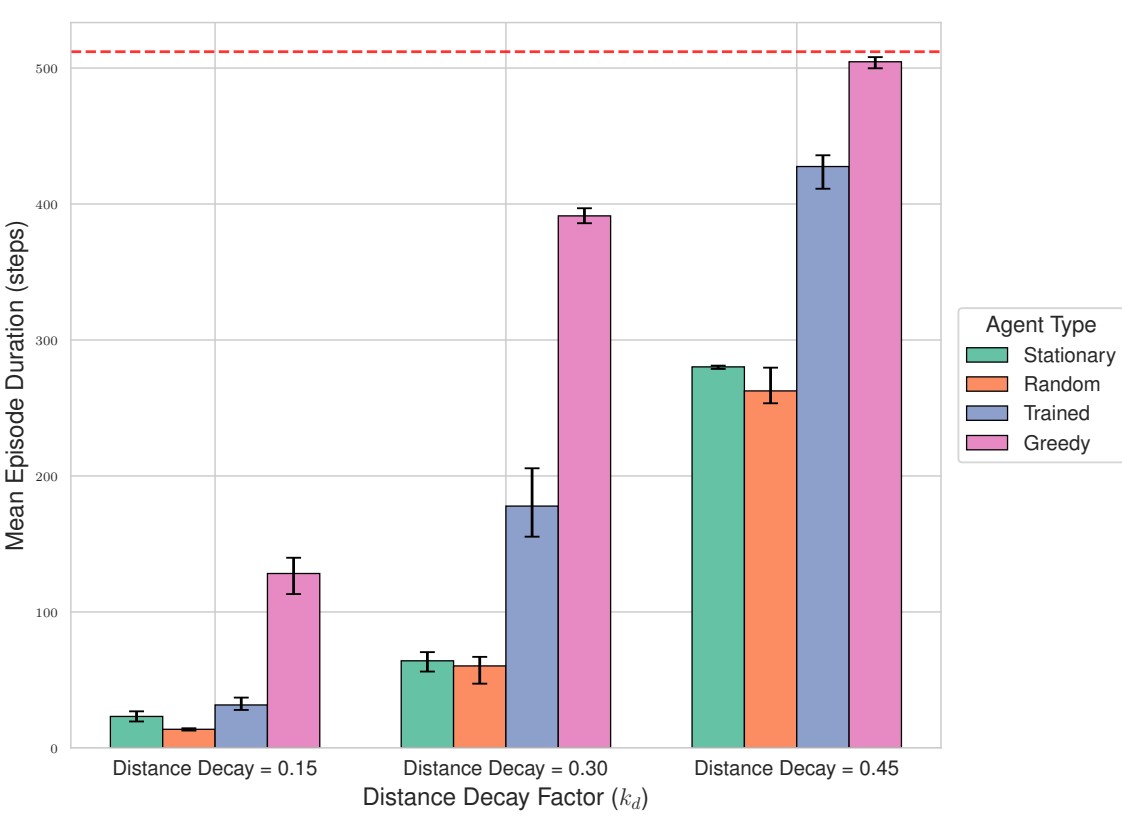

Figure 15: Performance comparison of *Trained* and baseline agents across varying values of the *distance decay* parameter, which controls how repulsion from infected individuals diminishes with spatial separation. Bars show mean episode durations, with 95% bootstrapped confidence intervals (10,000 samples) calculated from per-seed means. The red dashed line indicates the maximum episode duration. For a statistical analysis of whether *Trained* agents significantly outperform baselines, see Table 18.

