# OpenReview forum: "Reward Engineering for Spatial Epidemic Simulations: A Reinforcement Learning Platform for Individual Behavioral Learning"
_TMLR — Accepted by TMLR_

### Review · Reviewer_cDcn · 2025-12-10

**Summary Of Contributions:**

Recent works have studied epidemic simulations broadly within the context of Agent-Based Models (ABMs). These approaches apply heuristics as compared to general learning-based alternatives for learning fine-grained behavioral control. Towards this goal, the work presents ContagionRL, a Reinforcement Learning (RL) platform for studying spatial epidemic simulations through the lens of reward functions. Simulations define an agent which interacts with rule-based Gaussian humans and agents over a grid in order to survive and adhere to epidemic conditions. SIRS+D epidemiological model is employed for simulating dynamics with continuous control agents such as PPO, SAC and A2C considered in a continuous state and action space setup. At the core of the paper, the platform studies and integrates a range of reward functions (sparse to dense variants) in order to study the role of reward engineering in epidemic behavioral control. While sparse rewards such as Reduce Infection Prob are found to be effective, dense rewards such as Max Distance and the proposed Potential Field rewards provide a more informative signal. Evaluations assessing episode lengths and infections per timestep are conducted for various settings of rule-based agents, learned agents, reward functions, visibility radius, gride size and infection rate. Statistical tests demonstrate the suitability of Potential Field rewards along with the application of platform for studying agents corresponding to human movement patterns.

**Additional Comments:**

NA

**Audience:**

Yes

**Audience Explanation:**

The paper studies an important problem by both designing a platform for its simulation and illustrating the use of the platform using reward-based learning. From an application perspective, the work studies a practical problem receiving limited attention. From a theoretical and experimental standpoint, the work systematically dissects the contributions of the various components of the platform. I believe this is a detailed study in RL which might be of interest to both the machine learning and scientific community in general.

**Broader Impact Concerns:**

Authors should consider adding a broader impact statement discussing the significance of their work towards population-based studies and epidemic control from a public health perspective.

**Claims And Evidence:**

Yes

**Claims Explanation:**

* **Reward Function Dependency:** While the paper conducts a detailed analysis of the considered rewards, most reward functions characterize human-related pandemic behavior from a limited viewpoint. For instance, Alive and Reduce Infection Probability rewards, being sparse rewards, promote conservative and stagnant behavior. These functions simply force policies to stay away from infection. Note that from an epidemic viewpoint, this is the optimal policy. But from a social viewpoint, one would execute additional precautions or preparatory steps while moving closer to infection-prone regions. Trivially, sparse rewards (and their combinations) achieve optimal behavior by updating positions and moving away from risk-prone regions. That is, these functions emphasize more on the spatial information rather than the overall environment dynamics of the model.

* **Adherence:** Following on the reward function dependency, authors note that the spatial information is the more important aspect. However, it remains unclear as to how adherence, which is a more realistic and social aspect of epidemics, ties to the simulation. Most reward functions do not take into account adherence levels and treat these as stale state and action space information. It remains unclear as to how agents exploit the adherence information (or if at all they do) if the learning signal stems from spatial components of the domain. This limits an emphasis on the more practical aspects such as social restrictions, eg - social distancing, masks, sanitation, etc. Figure 11 compares different components of the Potential Field Reward. We observe that while spatial aspects such as direction and movement are important, adherence is also important in the overall design and optimization of the reward objective. However, Potential Field reward is the only objective that exploits this information. Thus, it remains unclear as to how adherence is utilized within the platform and what role does it play during the learning process.

* **Discrete RL:** Authors formulate the platform as a continuous control problem making use of common RL algorithms such as PPO, SAC and A2C. Additionally, heuristic baselines are also utilized to compare traditional rule-based approaches. It would be worth discussing the choice of the setup as a continuous control problem and how the platform could be customized or extended for formulating discrete problems. Similarly, authors could discuss algorithms (such as DQN, DDQN, Impala) which could be utilized within the epidemiological domains for fine-grained control.

* **Comparisons of Partial-Observability:** _"This counter-intuitive...observation spaces."_ Could the authors further explain this insight? It remains unclear as to how increased observation visibility may be attributed to state space noise or curse of dimensionality. In the case of noise, could the authors explain as to what is the importance of different components of the features present in the state representation? As in how is feature importance distributed across the state? This would help explain the role of different components of state features and how they benefit/deteriorate reward-based learning. Humans generally benefit from additional information during epidemics. Their actions become more deliberate, i.e- the policy distribution becomes sharper and peaks towards deterministic behaviors.

## Minors

* Page 6 last paragraph: of a each $\rightarrow$ of each
* Page 10 scond last paragraph: could you explain/study in detail the counter-intuitive result on limite visibility?
* Figure 6: 100 evaluat ion $\rightarrow$ evaluation
* Figure 6: Trained agent significantly $\rightarrow$ agents
* Figure 14: Could you explain the intuition and result of the plot? It is unclear as to how episode length should vary for different adherence levels.

**Requested Changes:**

Below are the changes critical for acceptance-
* **Adherence:** An analysis and discussion of adherence plays a key role in determining policy improvement towards epidemic rule compliance. Authors should study this component by taking into account 1-2 reward functions that take adherence into account. These functions can then be compared to Potential Field Reward and its various ablations. Additionally, authors should consider highlighting how agents learn policies corresponding to adherence. This can be achieved by showing that adherence, both in the state and action spaces, is adequately utilized and not simply dominated by spatial features.

Below are the changes that would strengthen the quality of work-
* **Movement Patterns:** Similar to the workplace movement pattern, authors could highlight or discuss additional movement patterns that are commonly observed during epidemics. For instance, home to grocery or home to clinic are common 2-location patterns. The platform could also cover/discuss additional 3-location patterns and qualitatively analyze the behavior of agents in simulation.

* **Discrete RL:** Customization in Rl environments has been a long-standing challenge. Adding support for discrete environments in the form of a customizable state and action space would be beneficial. Similarly, authors could highlight/discuss discrete RL algorithms that may be utilized in such cases.

* **Partial Observability:** Authors should consider explaining the counter-intuitive result on limited visibility. This can be achieved by studying feature importance ofthe state representation and perhaps providing more expressivity to RL agents in order to adapt to a larger dimensional state space.

---

> ### Author Response · Authors · 2026-01-12
> **Response to Reviewer cDcn (1/2)**
>
> We thank the reviewer for their thorough evaluation and constructive feedback. We appreciate the recognition of this work as a detailed RL study addressing an important and underexplored problem with practical applications to epidemic control. We also thank the reviewer for noting that our systematic experimental analysis is supported by accurate and convincing evidence. Below, we address the comments with new experiments and clarifications.
>
> ## Adherence (New Experiment)
> Our study demonstrates the agent learns to strategically maintain full adherence at decision points. However, its learnability depends on reward complexity.
>
> For the constant reward function that provides a reward of 1 for simply being alive, temporal credit assignment captures the significance of adherence. Similarly, for other reward functions **adherence can be learned implicitly** through survival signal shown by the mean adherence values when training with Constant, Reduce Infection, Constant + Reduce Inf, and Max Distance reward functions.
>
> While effective at capturing the importance of adherence, these reward functions cannot capture the implications of spatial navigation effectively and this led us to design the potential field reward function.
>
> In this experiment, we show that when adherence reward is removed from a complex multi-component objective like the potential field reward function (see PF: no adherence in Table 1), agents fail to discover its strategic value (essentially random usage with no correlation to infection risk). This proves adherence reward is not dominated by spatial features when using simple reward functions, but the opposite is true for complex reward functions.
>
> This is further supported from a game theory perspective, where setting the adherence cost to zero in training would mean that maximal adherence is the dominant strategy. ContagionRL platform supports a cost parameter for adherence for future exploration to understand the strategic trade-offs requiring adaptive adherence based on infection context.
>
>
> **Table 1: Learned Adherence Across Reward Functions**
>
> | Reward Function       | Mean Adherence | Std Adherence |
> | --------------------- | -------------- | ------------- |
> | Constant              | 1.00           | 0.00          |
> | Reduce Infection      | 1.00           | 0.00          |
> | Constant + Reduce Inf | 1.00           | 0.00          |
> | Max Distance          | 0.87           | 0.31          |
> | Potential Field       | 1.00           | 0.00          |
> | PF: No Adherence      | 0.41           | 0.47          |
> | PF: No Direction      | 1.00           | 0.00          |
> | PF: No Health         | 1.00           | 0.00          |
> | PF: No Magnitude      | 1.00           | 0.00          |
> | PF: No Move           | 1.00           | 0.00          |
> | PF: No S Repulsion    | 1.00           | 0.00          |
>
> ## Movement Patterns
> Our platform's modular `MovementHandler` class supports straightforward extension to additional patterns. The workplace-home cycle was chosen as a representative structured pattern that captures key epidemic features: temporal clustering, spatial segregation, and mixed structured/random movement. Implementing home-grocery or home-clinic patterns would simply require defining new spatial zones and visit schedules (e.g., grocery: 1-2 visits/week, clinic: conditional on infection). In Fig. 6 experiments, we showed that agents learn context-dependent strategies during workplace cycles through increased adherence during congregation periods and exploiting spatial dispersion during transitions. We expect similar patterns would emerge with grocery/clinic movements, creating weekly clustering cycles or state-dependent hotspots that agents must learn to navigate. While these extensions are valuable, we focused on a single structured pattern for controlled experimental conditions.

---

> ### Author Response · Authors · 2026-01-12
> **Response to Reviewer cDcn (2/2)**
>
> ## Discrete RL
> Our environment **supports discrete action spaces through Gymnasium's wrapper mechanism**.
>
> The current continuous action space can be discretized using a Gymnasium ActionWrapper that maps discrete actions to continuous values (e.g., 9 movement directions × 5 adherence levels). This approach requires zero modification to the core environment while enabling discrete RL algorithms such as DQN, QRDQN, and Rainbow as they are particularly suited for epidemic control due to their ability to handle sparse reward signals common in long-horizon intervention planning, in addition to the continuous algorithms (PPO, SAC, A2C) already demonstrated.
>
> The environment's modular design ensures discretization affects only action interpretation while preserving all epidemic dynamics. This flexibility allows researchers to compare continuous and discrete approaches on identical environment dynamics, which could be used by future work for studying policy representation trade-offs in epidemic control.
>
> ## Partial Observability (New Experiment)
> To address the counter-intuitive limited visibility result, we conducted a **feature importance analysis**, measuring how strongly each model responds to observed human positions. The table below shows Spearman correlations ($\rho$) between mean relative positions of visible humans and the agent's movement actions, denoted $\rho_{\Delta x}$ and $\rho_{\Delta y}$:
>
> **Table 2: Action-Position Correlation by Visibility Radius**
>
> | Visibility      | $\rho_{\Delta x}$ | $\rho_{\Delta y}$ |
> | --------------- | ----------------- | ----------------- |
> | Limited (r=10)  | -0.557            | -0.540            |
> | Limited (r=15)  | -0.441            | -0.464            |
> | Limited (r=20)  | -0.342            | -0.438            |
> | Full Visibility | -0.220            | -0.234            |
> The negative correlations indicate avoidance behavior (moving away from observed humans). Critically, **limited visibility models show 2-2.5× stronger correlations** (p<0.001), demonstrating that they respond more decisively to observed threats. This occurs because full visibility dilutes attention across 40 humans (many distant and irrelevant), while limited visibility filters the state space to only nearby relevant information.
>
> Importantly, this aligns with the infection mechanics: our environment uses distance decay $\beta_{\text{eff}} = \beta \times \exp(-0.3 \times d)$, meaning distant humans contribute negligibly to infection risk. Limited visibility naturally filters observations to the radius where humans actually pose significant threat, while full visibility forces the agent to process information about distant humans who have near-zero impact on infection probability. Rather than requiring more expressive architectures to handle larger state spaces, our results demonstrate that appropriately scoped observations matched to the task's distance-dependent mechanics produce more focused and effective policies.
>
> ## Broader Impact
> **We will add a brief broader impact section** discussing ContagionRL's potential to inform public health policy through simulated evaluation of behavioral interventions before real-world deployment.
>
> ## Fig. 14 Explanation
> The adherence effectiveness parameter $\alpha_{\text{eff}}$ controls NPI quality through the effective transmission rate: $\beta_{\text{eff}} = \beta(\alpha_{\text{eff}} + (1-\alpha_{\text{eff}})(1-\text{adherence}))$. Lower $\alpha_{\text{eff}}$ values represent higher-quality interventions (e.g., $\alpha_{\text{eff}}=0.2$ enables 80% risk reduction with perfect adherence). Episode length increases as $\alpha_{\text{eff}}$ decreases because more effective NPIs provide greater protection, allowing agents to survive longer before infection. The divergence between adherence levels at low $\alpha_{\text{eff}}$ demonstrates that high-quality NPIs amplify the value of strategic adherence.
>
> Thank you for pointing out the typographical errors. They will be corrected in the revision. Furthermore, Tables 1 and 2 will be added to the Appendix in the revised manuscript. Finally, we will **open-source the ContagionRL environment** and all accompanying code upon acceptance to support reproducibility and future research.

---

### Review · Reviewer_uG7X · 2025-12-14

**Summary Of Contributions:**

This paper proposes a framework for evaluating different individual-level behavioral strategies in epidemics. This is done in detail via a spatial agent-based model with dynamic interactions, i.e., agents move in space while potentially spreading the disease. It is assumed there is a single strategic agent who learns optimal policies for spatial movements via reinforcement learning (assuming an appropriate reward function is specified). The remaining agents move and interact in simple, natural ways. The authors then run extensive experiments in their framework, highlighting the impact of optimized decision-making, partial information and other properties.

Strengths:
The proposed framework is fairly detailed and flexible. I could see it being useful for testing various hypotheses and running counterfactual experiments. The authors also highlight an interesting result that if strategic agents act on partial information rather than full information, they may be a higher chance of staying uninfected for longer.

Weaknesses:
- While the system described in the paper is a good building block for more complex experiments, it is in my opinion too simplistic to draw serious insights. A large part of what makes epidemics interesting and unpredictable is that agent behaviors *collectively* propel the disease forward in a population. However, the focus here is only on a single agent, whereas a multi-agent RL setup may be more realistic.
- Given that key insights are drawn based on only one agent in a presumably large population, I wonder how necessary it is to simulate the entire large-scale system.

**Audience:**

Yes

**Audience Explanation:**

This paper studies an agent-based approach to a specific large-scale reinforcement learning problem. While the problem is highly specific and the framework is somewhat simplistic, it is a reasonable building block for more complex models and predictions.

**Broader Impact Concerns:**

I have no concerns related to the broader impacts of the work.

**Claims And Evidence:**

Yes

**Claims Explanation:**

The paper is clearly written, and all insights are supported by appropriate data and plots.

**Requested Changes:**

List of changes required for informing my decision:
- Please discuss how computational costs increase with the number of agents to simulate. Since we are only interested in the dynamics of one agent, I wonder if it is truly necessary to simulate the rest in detail, as this would be costly to compute. Perhaps a low-dimensional "fluid limit" could even be used in place of agent-based dynamics (except for the sole strategic agent). Please provide more context and information around these ideas.
- Please discuss how one would make this model more realistic, i.e., account for more detailed physical constraints and incorporate a multi-agent learning framework.
- Infections per timestep is a metric that is assessed in the paper. To my understanding, this captures the number of infections in the entire system per unit of time. If so, can you explain why this is used to assess the performance of a single strategic agent?

Minor changes / questions to be addressed:
- In your paper, $\alpha$ represents the adherence to an NPI. If this always reduces the likelihood of transmission, why wouldn't a strategic agent always choose the largest value it can be? If it is always advantageous to choose the largest value, why have it as an adjustable parameter?

---

> ### Author Response · Authors · 2026-01-12
> **Response to Reviewer uG7X**
>
> We thank the reviewer for their positive assessment. We appreciate the recognition of the framework's detail and flexibility for testing hypotheses and counterfactual experiments, the clarity of writing with insights supported by appropriate data, and the interesting nature of our partial observability findings. Below, we address the identified questions.
>
> ## Computational costs
> We conducted a computational scaling analysis to quantify how costs increase with population size.
>
> We found that per-step computation time scales as $O(n^{1.57})$, where $n$ is the number of agents. Our default configuration uses $n=40$. At larger populations, costs increase meaningfully. The scaling is driven primarily by pairwise infection probability calculations, which scale as $O(n × I)$ where $I$ is the infected count.
>
> **Table 1: Step Time Scaling with Agent Count**
>
> | Population Size (n) | Step Time (ms) | Steps/sec |
> | ------------------- | -------------- | --------- |
> | 40                  | 0.54           | 1,861     |
> | 80                  | 1.27           | 785       |
> | 160                 | 3.4            | 294       |
> | 320                 | 10.5           | 95        |
>
> Our individual-based formulation trades computational efficiency for modeling flexibility. While a fluid-limit approximation might reduce the infection calculation complexity, it would preclude the individual heterogeneity that our framework already leverages. Specifically, in Figure 10, we employ heterogeneous mobility patterns where 80% of agents follow individualized commute cycles between unique home and work locations, while 20% exhibit random movement, creating realistic urban mobility dynamics that it would be difficult to capture in a mean-field model.
>
> We agree with the reviewer that fluid-limit approximations represent a valuable direction for scaling to larger populations. Beyond the scope of the current paper, this formulation might evolve toward mean-field approximations for scalability or toward richer individual heterogeneity and landscape design for realism.
>
> ## Realism and Multi-Agent
> ContagionRL's modular architecture is designed to accommodate increasing physical realism. Natural extensions include enriching disease dynamics by modeling symptomatic/asymptomatic heterogeneity, which makes the RL problem more challenging. Mobility realism can be enhanced by constraining movement to network topologies inspired by activity-based movement, where agents follow more complex structured schedules among many locations (home, work, transit hubs). Our existing workplace/home cycle movement type shows this. Contact structure could incorporate duration-dependent transmission and location-specific factors (indoor/outdoor, crowding).
>
> Regarding multi-agent learning, understanding individual-level reward-behavior relationships is a necessary prerequisite for effective multi-agent design. Our ablation studies characterize what information and incentive structures an individual agent requires, and these findings transfer to multi-agent settings where each agent still learns from local rewards. Independent PPO (IPPO) to parameter-sharing variants for sample efficiency, enable study of emergent phenomena, including social distancing norm formation and free-rider dynamics. ContagionRL's observation space is amenable to multi-agent wrappers. We will expand the Discussion to outline these extension pathways.
>
> ## Infections per timestep
> Yes, infections per timestep is a system-level metric by design. We include it to characterize the epidemic conditions under which the strategic agent survives.
>
> Epidemic dynamics follow characteristic phases: growth (high transmission), peak, and decline (lower transmission). Different strategies yield different observation windows. Ineffective strategies succumb during high-transmission periods, sampling only the growth/peak phases for calculation of this metric. Effective strategies survive longer, with episodes spanning both high and low transmission phases.
>
> The lower infections per timestep for effective strategies reflects this averaging effect across the epidemic curve and not a causal reduction in transmission. This metric reveals that effective survival strategies enable agents to navigate through high-risk epidemic periods.
>
> ## Maximal NPI
> Maximum adherence reduces infection probability in our model (Eq. 2 in paper). Methodologically, allowing agents to _learn_ that maximal adherence is optimal (rather than hard-coding) is a simple validation.
>
> Our environment formulation includes an adherence penalty factor (Table 4, Appendix C.2) to capture these costs. This accommodates future work exploring how agents balance survival benefits against adherence costs under different penalty regimes.

---

### Review · Reviewer_oFM4 · 2025-12-28

**Summary Of Contributions:**

This paper develops a new platform for spatial epidemic simulations. The special feature of the new platform is that it encodes an agent
in the model, allowing its behavior to evolve over time upon interaction with the environment (while other individuals' behavior in the models is treated as static). On the platform, the agent learns the survival strategy via reinforcement learning algorithms, where different reward functions can be implemented. The paper provides numerical studies demonstrating what can be evaluated and compared
on the platform (e.g., different RL algorithms and different reward functions), drawing some preliminary conclusions based on the simulation results.

**Audience:**

Yes

**Audience Explanation:**

The idea of agent-based modeling in an epidemic simulation platform should be of interest to some audience.

**Broader Impact Concerns:**

NA.

**Claims And Evidence:**

No

**Claims Explanation:**

1. Although the authors provide some discussion of the rationale behind the single-agent design, it remains unclear why this simplification would not substantially affect the overall conclusions. In particular, it is not obvious that the qualitative insights would carry over unchanged. Could the authors comment on whether---and to what extent---the framework could be extended to a setting with a finite number of interacting agents?

2. The message of the paper appears to be a bit unclear to me. While the proposed ContagionRL framework is presented as a platform for evaluating and comparing different reinforcement-learning algorithms and reward functions, it is not evident how such comparisons are validated. For instance, if the single-agent formulation differs meaningfully from a setting in which each individual is modeled as an agent, the resulting conclusions could change. I wonder whether there is any notion of ground truth or external benchmark against which the proposed evaluations and comparisons are assessed.

**Requested Changes:**

Below are the questions/requested changes following up on my previous comments:

1. Could the authors comment on whether---and to what extent---the framework could be extended to a setting with a finite number of interacting agents?

2. Could the authors comment on whether there is any notion of ground truth or external benchmark against which the proposed evaluations and comparisons are assessed?

3. I wonder if the heterogeneity of the population can be encoded into the model. For example, different characteristics of the agent could lead to other behaviors and different model outcomes.

---

> ### Author Response · Authors · 2026-01-12
> **Response to Reviewer oFM4**
>
> We thank the reviewer for their thoughtful evaluation of our paper. We appreciate the recognition that agent-based modeling in epidemic simulation is of interest to the community. Below, we address the specific questions in turn.
>
> ## Extension to finite number of interacting agents
> The single-agent design is a deliberate methodological choice that allows us to focus on reward function design—the core scientific question of this work. Multi-agent reinforcement learning introduces confounding factors including non-stationarity from co-adapting agents, emergent social dynamics, and strategic interactions, all of which obscure whether observed behaviors stem from game-theoretic equilibria or from the reward structure itself. Given the black-box nature of deep RL training, disentangling these effects in a multi-agent setting would make it fundamentally unclear whether performance differences arise from reward engineering (our focus) or from emergent coordination/competition dynamics. By isolating a single learning agent within a controlled population, we can definitively identify which reward components—directional guidance, adherence incentives, potential field formulations—are critical for effective policy learning at the individual level.
>
> Importantly, understanding individual-level reward-behavior relationships is a necessary prerequisite for designing effective reward systems in multi-agent deployments. The insights from our ablation studies (e.g., that directional cues and explicit adherence rewards are essential while movement magnitude is not) characterize what information and incentive structures an individual agent requires to learn risk-averse behavior. These findings about reward component necessity are likely to transfer to multi-agent settings, where each agent still faces the fundamental challenge of learning from its local reward signal.
>
> We view our work as establishing the foundational understanding that must precede principled multi-agent extensions. Looking forward, ContagionRL's modular architecture is amenable to such extensions: For instance, incorporating Independent PPO (IPPO) where each agent maintains its own policy network while sharing the environment, implemented via wrapper classes, with optional parameter sharing to reduce sample complexity. We will clarify this methodological rationale and discuss multi-agent extension pathways in the revised manuscript.
>
> ## Ground truth or external benchmark
> Our work addresses a novel problem formulation: training a single agent to learn both spatial navigation and NPI adherence decisions in response to localized epidemic conditions. No established benchmark exists that can jointly evaluate both dimensions of this dual-action learning problem.
>
> Adjacent benchmarks address only partial aspects. Social robot navigation benchmarks (CrowdNav, SocNavBench) evaluate spatial navigation among pedestrians but lack disease transmission dynamics or protective behavioral choices. Conversely, epidemic RL benchmarks (gym_epidemic, SIRV-B) train policy-maker agents controlling population-level interventions without spatial navigation. Individual navigation through an epidemic with personal protective decisions seems to be a relatively unexplored model system.
>
> Given that no unified benchmark exists, we have conducted comparisons against interpretable baselines with statistical significance testing across multiple independent runs.
>
> ## Heterogeneity in Modelling
> Our platform currently implements heterogeneity at the individual level in the form of movement pattern differentiation. In our workplace/home cycle experiments (Figure 6), we explicitly stratify the population into two behavioral subgroups: Humans who follow a structured commute pattern and others who exhibit continuous random movement with momentum-based trajectories. Additionally, our `MovementHandler` tracks per-individual velocity vectors, enabling each human's trajectory to evolve independently based on their personal movement history.
>
> The platform's architecture can support extension to richer forms of heterogeneity, for instance including age-stratified populations, variable immune system strength, differential susceptibility, heterogeneous recovery rates, and varied mortality risk. Our 20+ configurable parameter system (Table 4 in main paper) provides the infrastructure for these extensions. Furthermore, the framework could be extended to multiple learning agents with distinct characteristics. This would allow future study of how heterogeneous adaptive behaviors emerge across different demographic or risk groups. We agree that introducing characteristics such as age-dependent mortality, asymptomatic carriers with reduced transmission, incubation periods and varied NPI adherence costs across subpopulations would create valuable future research directions.

---

### Decision · Action_Editor_bbiU · 2026-02-15

**Recommendation:** Accept with minor revision

**Additional Comments:**

The AC felt that a few things should have been made clearer for better readability.

1. It is not clear how different reward terms are computed. For example, for the directional term, does it depend on the local information, or is some global information used? Some discussions would help the reader.

2. While the single-agent setting is fine, as many reviewers have pointed out, the impact on the multi-agent interaction system would be important. Hence, few metrics such as total infections caused/avoided on the population-level would make the paper more complete.

**Audience:**

Yes

**Audience Explanation:**

The contributions would be important to the TMLR community.

**Claims And Evidence:**

Yes

**Claims Explanation:**

The paper introduces ContagionRL, a grid-based SIRS+D epidemic simulator with a single controllable RL agent embedded in a population of non-learning humans. The agent controls movement and NPI adherence, and the environment supports both MDP and POMDP.  The core scientific claim is that reward design materially determines whether RL learns meaningful “infection-avoidance navigation.” The authors compare several reward formulations and propose a Potential-Field reward. They include nonparametric significance tests and an ablation study identifying which Potential-Field components matter. The main contribution of the paper is to develop a simulator-based testbed for RL-agent. The paper has conducted an extensive study of the effects of reward structure. The paper has provided a key and interesting insight. Overall, according to the AC,the claims made in the paper are **mostly** convincing and well justified. The AC has provided some feedback to the authors on how to make the paper more complete. The AC has thus recommended Acceptance with Minor revision.